# Impact of iron fertilisation on atmospheric $CO_2$ during the last glaciation

Himadri Saini[1,2], Katrin J. Meissner[1,2], Laurie Menviel[1,3], and Karin Kvale[4]

[1]Climate Change Research Centre, University of New South Wales, Sydney, New South Wales, Australia
[2]The Australian Research Council Centre of Excellence for Climate Extremes, Sydney, New South Wales, Australia
[3]The Australian Centre for Excellence in Antarctic Science, University of Tasmania, Hobart, Tasmania 7001, Australia
[4]GNS Science, 1 Fairway Drive, Avalon 5010, P.O. Box 30368, Lower Hutt 5040, New Zealand

**Correspondence:** Himadri Saini (himadri.saini@student.unsw.edu.au)

**Abstract.** While several processes have been identified to explain the decrease in atmospheric $CO_2$ during glaciations, a better quantification of the contribution of each of these processes is needed. For example, enhanced aeolian iron input into the ocean during glacial times has been suggested to drive a 5 to 28 ppm atmospheric $CO_2$ decrease. Here, we constrain this contribution by performing a set of sensitivity experiments with different aeolian iron input patterns and iron solubility factors under boundary conditions corresponding to 70 thousand years before present (70 ka BP), a time period characterised by the first observed peak in glacial dust flux. We show that the decrease in $CO_2$ as a function of Southern Ocean iron input follows an exponential decay relationship. This exponential decay response arises due to the saturation of the biological pump efficiency and levels out at ∼21 ppm in our simulations. We show that the changes in atmospheric $CO_2$ are more sensitive to the solubility of iron in the ocean, than the regional distribution of the iron fluxes. If surface water iron solubility is considered constant through time, we find a $CO_2$ draw-down of ∼4 to ∼8 ppm. However, there is evidence that iron solubility was higher during glacial times. A best estimate of solubility changing from 1% during interglacials to 3 to 5% under glacial conditions yields a ∼9 to 11 ppm $CO_2$ decrease at 70 ka BP, while a plausible range of $CO_2$ draw-down between 4 to 16 ppm is obtained using the wider but possible range of 1 to 10%. This would account for ∼12-50% of the reconstructed decrease in atmospheric $CO_2$ (∼32 ppm) between 71 and 64 ka BP. We further find that in our simulations the decrease in atmospheric $CO_2$ concentration is solely driven by iron fluxes south of the Antarctic polar front, while iron fertilization elsewhere plays a negligible role.

## 1 Introduction

$CO_2$ draw-down during the last glacial period occurred in multiple steps before reaching a minimum level of ∼190 ppm at the Last Glacial Maximum (LGM, 21,000 years ago) (EPICA Community Members et al., 2004; Ahn and Brook, 2008; Lüthi et al., 2008; Bereiter et al., 2012, 2015). One of the last major drops in atmospheric $CO_2$ concentration occurred at the beginning of Marine Isotope Stage (MIS) 4 (71-59 thousand years ago, ka BP hereafter) when $CO_2$ decreased from 233±8 ppm to 201±4 between 71 ka and 64 ka (Figure 1b). This period also coincided with the minimum summer insolation at high northern latitudes (∼70 ka BP, Berger (1978)), which led to the most pronounced episode of ice sheet growth in the Northern Hemisphere (NH) over the glaciation (Bassinot et al., 1994; Petit et al., 1999; Grant et al., 2012). Even though NH ice sheet extent and volume

during MIS4 and MIS3 are not well constrained, numerical modelling experiments and observational records have shown that the extent of the Cordilleran and Scandinavian ice sheets was larger during MIS4 than during the LGM, whereas the Laurentide and Greenland ice sheets were smaller (Lambeck et al., 2010; Kleman et al., 2013; Batchelor et al., 2019). As a result of the glaciation occurring during the early part of MIS4, it is suggested that the global sea level dropped to about 80m below present day sea level (Waelbroeck et al., 2002; Grant et al., 2012; Batchelor et al., 2019; De Deckker et al., 2019). Global sea surface temperatures (SSTs) also show a rapid decline by $\sim$1°C in the early MIS4 transition and an overall $\sim$1.5° drop across MIS4, before rising again to pre-MIS4 levels around 59 ka BP (Kohfeld and Chase, 2017).

Several mechanisms have been put forward to explain the draw-down of atmospheric $CO_2$ during glaciations, including the $\sim$32 ppm drop in $CO_2$ during the MIS4 transition. These mechanisms include higher solubility of $CO_2$ in colder ocean waters (Heinze et al., 1991; Kucera et al., 2005; Williams and Follows, 2011; Khatiwala et al., 2019); higher carbon sequestration associated with a weaker Atlantic Meridional Overturning Circulation (AMOC) and thus lower ventilation rates during glacial periods (Sigman and Boyle, 2000; Toggweiler, 2008; Sigman et al., 2010; Watson et al., 2015; Jaccard et al., 2016; Yu et al., 2016; Menviel et al., 2017; Kohfeld and Chase, 2017); and the expansion of sea ice cover leading to more stratified Southern Ocean (SO) waters and smaller air-sea gas exchange (Francois et al., 1997; Stephens and Keeling, 2000; Ferrari et al., 2014). However, it has been argued that sea ice expansion does not correlate well with the timing of $CO_2$ draw-down at 70 ka (Kohfeld and Chase, 2017), and that the major drivers during this transition might in fact be due to a shallower AMOC (Piotrowski et al., 2005; Thornalley et al., 2013; Yu et al., 2016), or a more efficient biological pump due to enhanced iron input to the ocean (Brovkin et al., 2012; Menviel et al., 2012; Anderson et al., 2014; Lamy et al., 2014; Martínez-García et al., 2014). Interestingly, high resolution $\delta^{13}CO_2$ records from Antarctic ice cores (Menking et al., 2022) display a 0.5 permil decrease centred at 70.5 ka, followed by a 0.7 permil increase (Figure 1g), indicating a complex set of processes impacting atmospheric $CO_2$ at the MIS5-4 transition. While surface ocean cooling could explain the concurrent $CO_2$ and $\delta^{13}CO_2$ decrease, the $\delta^{13}CO_2$ increase in the second part of the transition would be consistent with a greater efficiency of the biological pump and increased storage of respired carbon in the deep ocean (Menviel et al., 2015; Eggleston et al., 2016; Menking et al., 2022).

The efficiency of the biological pump is partly dependent on the relative abundance of different marine phytoplankton communities, which further depends on the availability of both macro and micro nutrients (Kvale et al., 2015a; Saini et al., 2021). Martin (1990) suggested that the micro nutrient iron plays a crucial role as a limiting nutrient in phytoplankton growth, and thus put forward the hypothesis that iron fertilisation, resulting from an increase in atmospheric dust during glacial times, might have increased marine net primary production (NPP) in the Southern Ocean, leading to a decrease in atmospheric $CO_2$ concentration. Antarctic ice core records indeed show peaks in dust fluxes that coincide with lower Antarctic temperatures and lower $CO_2$ levels (Figure 1) during MIS4 and MIS2 (27-19 ka) (Wolff et al., 2006, 2010; Lambert et al., 2008, 2012; Martínez-García et al., 2011, 2014; Lamy et al., 2014). These peaks in dust flux are concurrent with increased export production (EP) in the subantarctic zone (SAZ) of the Southern Ocean during the LGM (Kohfeld et al., 2005, 2013; Martínez-García et al., 2014), as well as during MIS4 (Lamy et al., 2014; Martínez-García et al., 2014; Thöle et al., 2019; Amsler et al., 2022). On the other hand, palaeoceanographic data from the Antarctic zone (AZ) suggest a decrease in EP at the LGM and MIS4 (Kohfeld et al.,

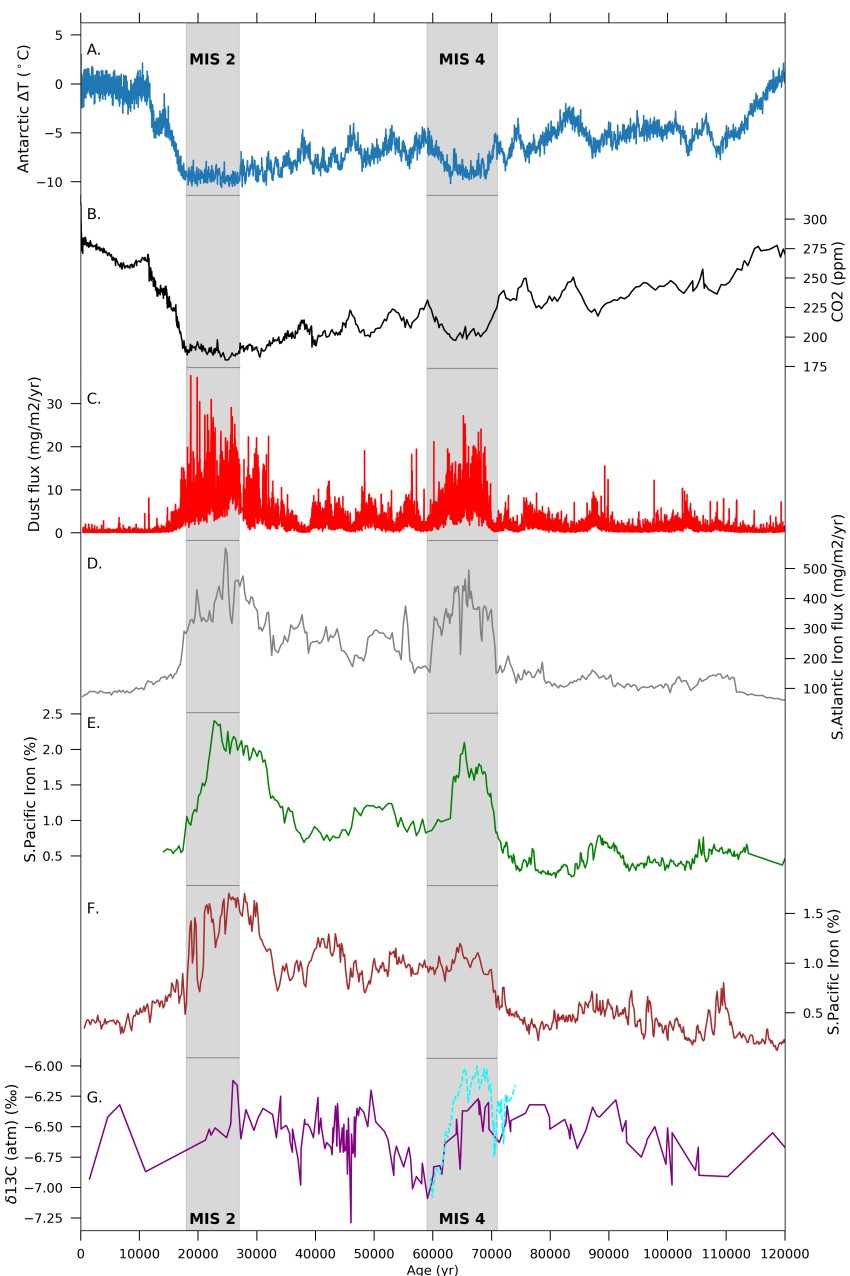

**Figure 1.** Time series of (a) Antarctic temperature anomalies from present day (°C) (Jouzel et al., 2007), (b) atmospheric $CO_2$ concentration (ppm) (Bereiter et al., 2015), and (c) dust flux (mg/m2/yr) (Lambert et al., 2012) as recorded in EPICA DOME C ice core; (d) iron dust accumulation rates (mg/m$^2$/yr) from ODP Site 177-1090 (Atlantic) (Martínez-García et al., 2011); iron (%) records in the South Pacific from Lamy et al. (2014) at sites (e) PS75-076 and (f) PS75-059; (g) atmospheric $\delta^{13}CO_2$ (‰) as recorded in a composite of Antarctic ice cores (purple line, Eggleston et al. (2016)) and high resolution records from Taylor Glacier ice cores (dashed cyan line, Menking et al. (2022)). Shaded areas represent marine isotope stages 2 (MIS2: 27-19 ka BP) and 4 (MIS4: 71-59 ka BP).

2005, 2013; Anderson et al., 2009; Jaccard et al., 2013; Studer et al., 2015; Thöle et al., 2019; Amsler et al., 2022; Weber et al., 2022).

Numerous modelling studies have explored the changes in Southern Ocean biogeochemistry due to enhanced aeolian iron input either under pre-industrial (PI) (Aumont and Bopp, 2006; Tagliabue et al., 2009a, 2014) or LGM conditions (Aumont et al., 2003; Bopp et al., 2003; Oka et al., 2011; Lambert et al., 2015; Muglia et al., 2017; Khatiwala et al., 2019; Yamamoto et al., 2019; Lambert et al., 2021; Saini et al., 2021), but not under MIS4 conditions. An exception is Menviel et al. (2012) who simulated a $\sim$12 ppm $CO_2$ drop as a result of enhanced aeolian iron fluxes into the Southern Ocean at 70 ka in a transient

simulation of the last glacial cycle.

In this study, we constrain the impact of enhanced aeolian iron input on atmospheric $CO_2$ concentration during the MIS4 transition. We perform a set of sensitivity experiments with different aeolian iron flux masks and different iron solubilities under 70 ka boundary conditions, using a model of intermediate complexity, which includes an ocean general circulation model (OGCM) coupled to a recently developed complex ecosystem model (Kvale et al., 2021; Saini et al., 2021). The ecosystem

model used here includes calcifying and silicifying plankton and an iron cycle, which makes it well-suited for studying $CO_2$ uptake in the Southern Ocean.

## 2    Methods

### 2.1    Model description

This study uses the 2.9 version of the University of Victoria Earth System Climate Model (UVic ESCM), which consists of a

sea ice model (Semtner Jr, 1976; Hibler, 1979; Hunke and Dukowicz, 1997) coupled to the ocean general circulation model MOM2 (Pacanowski, 1995), with 19 ocean depth layers and a spatial resolution of 3.6° by 1.8°. It also includes a dynamic vegetation model (Meissner et al., 2003), a land surface scheme (Meissner et al., 2003), a 2-D atmospheric energy moisture balance model integrated vertically (Fanning and Weaver, 1996) and a sediment model (Archer and Maier-Reimer, 1994; Meissner et al., 2012). The model is forced with seasonally varying wind stress and wind fields (Kalnay et al., 1996) and

seasonal variations in solar insolation at the top of the atmosphere. Full physical and structural descriptions of the model can be found in Weaver et al. (2001), Meissner et al. (2003), Eby et al. (2009), and Mengis et al. (2020).

The marine carbon cycle is represented by the newly developed Kiel Marine Biogeochemistry Model, version 3 (KMBM3) (Kvale et al., 2021) which is based on the Nutrient Phytoplankton Detritus Zooplankton model of Schmittner et al. (2005) and Keller et al. (2012). There are four different classes for phytoplankton in this ecosystem model. Three of which include

specifically characterized plankton such as diazotrophs that can fix nitrogen, coccolithophores that produce $CaCO_3$ shells, and diatoms that produce opal. The fourth class is for the rest of the types of plankton, that are mostly located in the low latitude regions. The model also includes prognostic $CaCO_3$ and silica tracers (Kvale et al., 2015b, a, 2021), dissolved nitrate, phosphate, iron, and silica as nutrients. The model also incorporates an iron cycle (Nickelsen et al., 2015), including hydrothermal sources. The ecosystem model is described in detail in Kvale et al. (2021).

**Table 1.** Aeolian iron fluxes for each experiment.

| Experiment | Total iron flux into the ocean $(Gmol yr^{-1})$ | Iron flux into the Southern Ocean south of 47°S $(Gmol yr^{-1})$ |
|---|---|---|
| PI-control (pife1%) | 3.47 | 0.092 |
| 70ka-control (pife1%) | 3.47 | 0.092 |
| lambfe1% | 9.072 | 0.461 |
| lambfe3% | 27.44 | 1.383 |
| lambfe5% | 45.73 | 2.305 |
| lambfe7% | 64.02 | 3.227 |
| lambfe10% | 91.46 | 4.610 |
| lambfe20% | 182.9 | 9.221 |
| glacfe1% | 6.532 | 0.4425 |
| glacfe3% | 19.6 | 1.328 |
| glacfe10% | 65.32 | 4.425 |
| glacfe20% | 139.1 | 8.850 |
| lambfe10%-30S | 18.51 | 4.610 |
| lambfe10%-40S | 13.94 | 4.610 |
| lambfe10%-50S | 6.05 | 2.66 |
| lambfe10%-60S | 3.9 | 0.5 |
| lambfe50%-47S | 27.09 | 23.05 |
| pife3% | 10.41 | 0.27 |
| pife5% | 17.35 | 0.46 |
| pife7% | 24.29 | 0.64 |
| pife10% | 34.7 | 0.92 |
| pife20% | 69.4 | 1.84 |

## 2.2 Experimental design

A pre-industrial control simulation (PI-control) is integrated under PI boundary conditions, including an atmospheric $CO_2$ concentration of 283.86 ppm, and PI iron (pife) and silica (pisi) fluxes. The PI aeolian dust fluxes are based on the PI-BASE dust flux simulation of Mahowald et al. (2006). To obtain iron and silica fluxes, we multiply the dust flux with the percentage distribution of iron and silica in dust (Zhang et al., 2015), respectively. The solubility factor for iron in the PI-control simulation is set to 1%.

All the other experiments are run under 70 ka boundary conditions, including orbital parameters corresponding to the year 70 ka BP according to Berger (1978), and a continental ice sheet topography generated by an offline ice-sheet simulation performed with IcIES (Ice sheet model for Integrated Earth system Studies, Abe-Ouchi et al. (2013)). The global mean ocean alkalinity was adjusted on the basis of sea level differences. For the control simulation (70ka-control), the model is run into

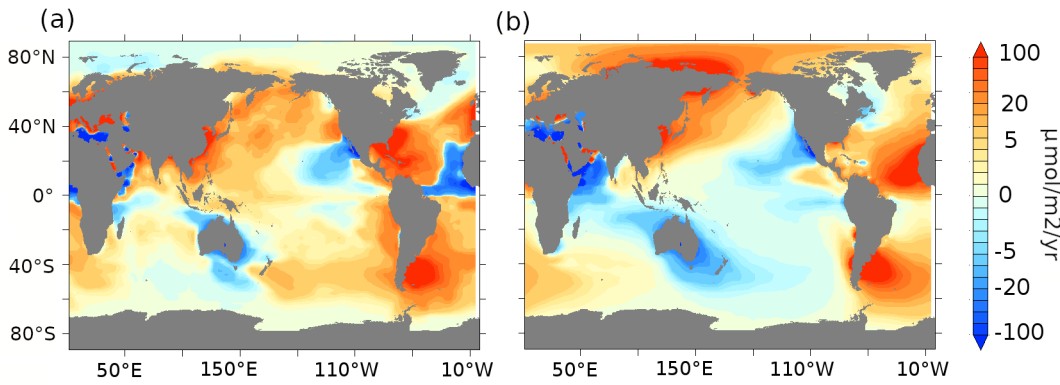

**Figure 2.** Aeolian iron dust flux ($\mu$mol m$^{-2}$ yr$^{-1}$) anomalies for (a) lambfe1% minus pife1% and (b) glacfe1% minus pife1%.

equilibrium with an atmospheric $CO_2$ concentration of 222.5 ppm (Bereiter et al., 2015), after which the simulation is integrated for an additional 200 years with prognostic atmospheric $CO_2$. 70ka-control is forced with pife (with 1% iron solubility) and pisi fluxes.

From this control simulation, we branch off a suite of sensitivity experiments with prognostic atmospheric $CO_2$, using two different glacial iron dust flux estimates. The first iron flux, lambfe, is obtained from the LGM dust estimates of Lambert et al. (2015). This dust flux estimate is calculated by performing a global 2-d interpolation on unevenly distributed LGM dust flux records, most of which are collated in the DIRTMAP (Dust Indicators and Records of Terrestrial and Marine Paleoenvironments) database (Kohfeld and Harrison, 2001; Maher et al., 2010). Their interpolation method assumes that the aerosol concentration in the air decreases exponentially from the source distance. The second iron flux, glacfe, is derived from a dust flux obtained with a model (LGMglac.a, Ohgaito et al. (2018). This model includes glaciogenic dust sources in addition to the usual desert dust sources and assumes dry and unvegetated regions as dust sources. It then calculates global dust transport and deposition. Both glacfe and lambfe are then obtained in this study by mapping the iron percentage on dust (Zhang et al., 2015) as mentioned above. The aeolian iron fluxes based on glacfe and lambfe dust patterns compared to the PI iron flux (pife), and assuming a 1% solubility, are shown in Figure 2, while the full dust deposition maps for PI and the two LGM reconstructions are available in the supplementary material of Saini et al. (2021).

The iron delivered to the ocean surface via aeolian dust deposition needs to be dissolved before it can become available to phytoplankton. The solubility factor of iron is therefore an important parameter that will impact the magnitude of iron fertilization. Iron solubility varies depending on both provenance and speciation. For example, Baker et al. (2006) find that solubilities in samples of Saharan dust (median of 1.7%) are significantly lower than solubilities in aerosols from other source regions feeding into the Atlantic ocean (median of 5.2%). Schroth et al. (2009) find a significant difference in iron solubility between arid soils (1%), glacial flour (2-3%) and, of lesser importance to our study, oil combustion products (77-81%). In the present day Southern Ocean, observations of iron solubilities vary between 0.2 and 48% (Ito et al., 2019), with the higher values being hypothesized to be caused by pyrogenic iron. The majority of the observational data in the present day Southern Ocean

lies between 1 and 12% (Ito et al., 2019). Quantifying iron solubilities during past climates is even more challenging. Conway et al. (2015) estimate iron solubility at the Last Glacial Maximum (LGM) based on dust particles in the EPICA Dome C and Berkner Island ice cores. They find that iron solubility was very variable during the LGM interval at Dome C (1-42%), with a mean of 10% and a median of 6%. On the other hand, iron solubility at Berkner Island was 1-5% at the LGM, and in average $\sim$3% between 23 and 50 ka. The lower iron solubility at Berkner Island is thought to be due to the aerosol composition. Due to its proximity to South America, the solubility found at Berkner island might better represent large-scale aeolian dust deposited in the Southern Ocean. Iron is more bioavailable in dust that originates from physically weathered than from chemically weathered bedrock (Shoenfelt et al., 2017). The analysis of subantarctic marine sediment cores further suggests that aeolian iron was 15 to 20 times more bioavailable during glacial periods than during the current interglacial (Shoenfelt et al., 2018).

In this study, we test a global mean solubility range of 1% to 20% (Table 1). Based on the studies mentioned above, we define the plausible range of average iron solubility to be 1 to 10%, with an increased likelihood at 3-5% during colder climate episodes. During warmer climates, such as PI or at the MIS4 onset, we define the most likely solubility factor as 1%.

However, due to the uncertainties associated with present-day iron solubilities, we perform additional sensitivity experiments under 70 ka BP boundary conditions using the PI iron dust mask with iron solubility varying between 3% and 20%. This approach allows us to estimate the minimum change in $CO_2$ due to glacial dust fluxes, assuming no change in solubility over time. The corresponding $CO_2$ changes can be calculated by taking the difference between $CO_2$ changes achieved with the full experiments (i.e., changing masks and solubilities) and the $CO_2$ changes achieved by only changing solubility. This approach was validated by performing two additional 70ka equilibrium experiments with the pife mask and an iron solubility of 3% and 10% from which we branched off simulations with the lambfe mask and constant solubility (not shown). The resulting $CO_2$ drawdown in these experiments was the same than if calculated as the difference between the full experiments and solubility-only experiments. The integrated aeolian iron input for all experiments is listed in Table 1.

We perform five additional sensitivity experiments (Figure A1) to better quantify the contribution of the Southern Ocean to the total $CO_2$ draw-down. In these experiments, the aeolian iron flux in the Southern Ocean follows the lambfe mask while the PI aeolian iron fluxes are applied outside of the Southern Ocean. In four of these experiments (lambfe10%-30S, lambfe10%-40S, lambfe10%-50S, lambfe10%-60S), the lambfe mask with 10% solubility (Figure A2) is applied south of 30°S, 40°S, 50°S and 60°S, respectively. In the fifth sensitivity experiment (lambfe50%-47S), the aeolian iron input south of 47°S follows the lambfe mask with a solubility factor of 50% (Figure A1f), which is equivalent to 23.05 Gmolyr$^{-1}$ iron input in the Southern Ocean south of 47°S and provides an upper limit on the potential $CO_2$ draw-down.

## 2.3 Carbon decomposition

To better understand the changes in ocean carbon, we decompose the simulated dissolved inorganic carbon (DIC) into its three major components: respired organic carbon ($C_{reg}$), DIC generated by dissolution of calcium carbonate ($C_{CaCO_3}$) and preformed carbon ($C_{pref}$) as described below.

$C_{reg}$ is calculated based on the remineralized phosphate in the ocean ($P_{reg}$) and the carbon to phosphate stoichiometric ratio ($R_{C/P}$=106):

$$\Delta C_{reg} = \Delta P_{reg} \times R_{C/P} \qquad (1)$$

$P_{reg}$ is determined based on Apparent Oxygen Utilisation (AOU) and the phosphate to oxygen stoichiometric ratio ($R_{P/O2}$=1/160), $\Delta P_{reg} = \Delta AOU \times R_{P/O2}$, where AOU is calculated as the difference between the saturated oxygen concentration ($O_{2sat}$) and the dissolved oxygen concentration in the ocean ($O_2$); $AOU = O_{2sat} - O_2$.

$C_{CaCO_3}$ is calculated as:

$$\Delta C_{CaCO_3} = 0.5(\Delta ALK + R_{N/P} \times \Delta P_{tot}) \qquad (2)$$

where $R_{N/P}$= 16. And finally, $C_{pref}$ is calculated as the remainder:

$$\Delta C_{pref} = \Delta DIC - \Delta C_{reg} - \Delta C_{CaCO_3} \qquad (3)$$

To further assess the efficiency of the biological pump, we calculate global P* (Ito and Follows, 2005), defined as:

$$P^* = P_{reg}/P_{total} \qquad (4)$$

where $P_{total}$ is the total phosphate content in the ocean.

## 3 Results

### 3.1 Simulated ocean conditions at 70 ka BP

In this section, we describe the simulated physical and biological conditions at 70 ka BP, at the onset of MIS4 (Figure 3). In our 70ka-control simulation, the globally averaged ocean temperature is ~2.8°C, about 0.6°C lower than in PI-control, but 0.9°C higher than in a LGM simulation integrated with the same model (Saini et al., 2021). The globally averaged annual mean SST is 17.1°C, 0.8 degrees lower than in the PI-control and 1.1 degrees higher than in the LGM simulation. The strongest ocean surface cooling (-1.45°C) at 70 ka BP with respect to our PI-control is simulated north of 40°N in the Atlantic and Pacific oceans, while the annual mean SSTs in the SAZ and AZ are ~0.8°C and 0.4°C lower than in the PI-control, respectively (Figure 3a). At 70 ka BP, the annual mean Southern Ocean sea ice edge is situated at 55°S in the South Atlantic and the South Indian Ocean, ~1 degree further north than in the PI-control simulation, while the change is insignificant in the South Pacific sector (sea ice edge at ~62°S). The simulated AMOC strength in the 70ka-control experiment is ~14 Sv, compared to ~17 Sv in the PI-control, without significant changes in its depth (Figure A3). Furthermore, there are no significant changes in simulated Antarctic Bottom Water (AABW) formation (Figure 3c, d).

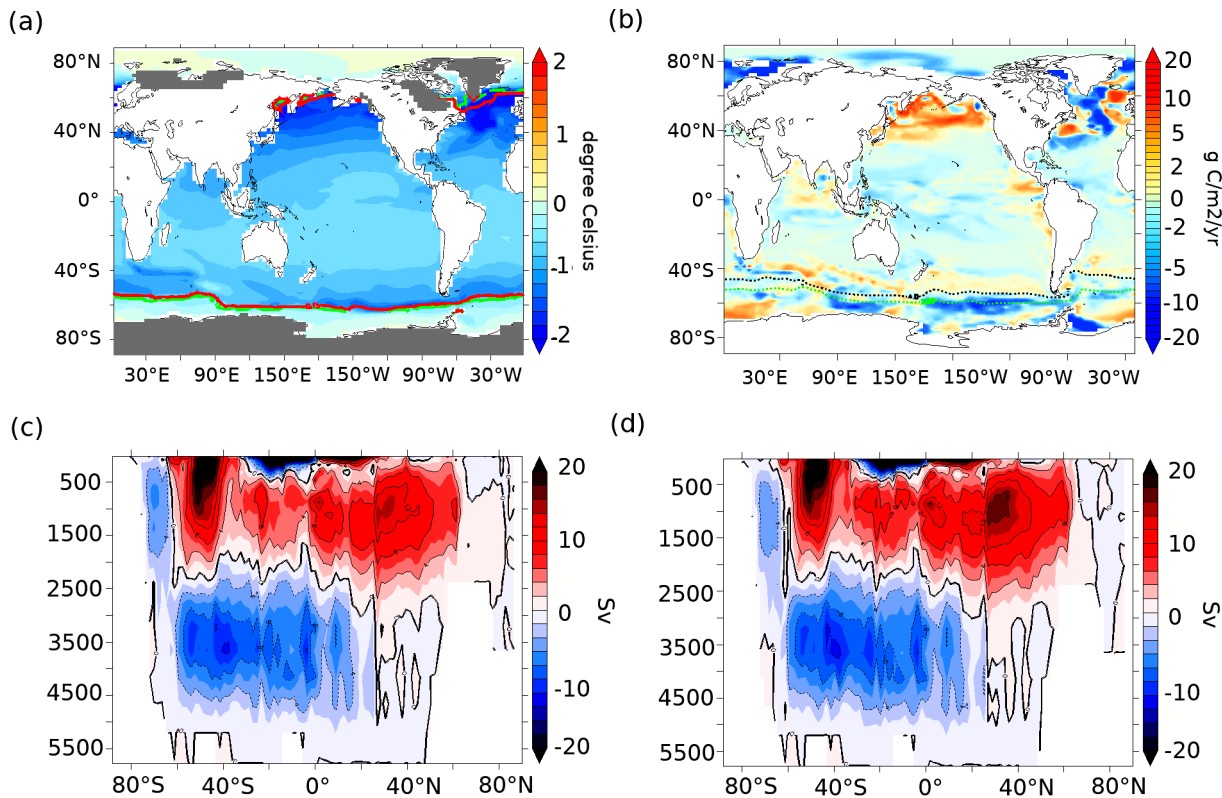

**Figure 3.** (a) Annual mean SST anomalies (°C, shading) at 70 ka BP compared to PI and 15% annual mean sea-ice concentration at PI (green line) and 70 ka BP (red line). The grey shading on land shows the extent of continental ice sheets at 70 ka BP (Abe-Ouchi et al., 2013); (b) Export production anomalies between 70 ka BP and PI at 177.5m depth (gC m$^{-2}$ yr$^{-1}$). Overlaid dashed contours represents modern subantarctic front (SAF) in black and Antarctic polar front (APF) in green based on Sokolov and Rintoul (2009); Global overturning streamfunction (Sv) at (c) 70 ka BP and (d) PI.

The colder conditions, more extensive sea-ice cover, and changes in the deep ocean convection impact marine productivity (Saini et al., 2021). Globally, NPP decreases by ∼9.7%, while EP decreases by ∼3% in our 70ka-control simulation compared to PI-control (both integrated with the pife mask and 1% iron solubility, Table A1). The simulated 18% EP increase at 70 ka in the North Pacific (Figure 3b) can be attributed to a greater diatom and coccolithophore abundance in that region (Figure A4a, b) resulting from higher nutrient availability. This EP increase in the North Pacific is however inconsistent with the biogenic Ba (Jaccard et al., 2005) and $\delta^{15}$N (Gebhardt et al., 2008) records from the sub-arctic North Pacific which suggest a decrease in EP at MIS4 onset. The North Atlantic shows a complex pattern of anomalies due to changes in the strength and location of deep ocean convection, which result in an overall decrease in NPP and EP by 16% and 9%, respectively.

Within the AZ (south of the APF), diatoms decrease by 14% in the Pacific, while they increase by 3% in the Atlantic and Indian sectors (Figure A4a). The decrease in diatoms in the Pacific sector of the AZ leads to a competitive growth advantage

**Table 2.** $\Delta CO_2$ simulated by changing the iron mask from PI to glacial (lambfe and glacfe) at 70 ka BP but keeping the solubility constant.

| Solubility | $\Delta CO_2$ (lambfe-pife) | $\Delta CO_2$ (glacfe-pife) |
|---|---|---|
| 1% | -3.8 | -5.1 |
| 3% | -6.4 | -8.2 |
| 5% | -6.9 | - |
| 7% | -7.5 | - |
| 10% | -7.3 | -8.3 |
| 20% | -6.7 | -6.9 |

for coccolithophores, which increase by 6.5% south of the polar front, thus leading to a poleward shift of coccolithophores in the Pacific sector (Figure A4b). On the other hand, coccolithophores decrease in the Atlantic and Indian sectors of the AZ by 15% and 8%, respectively. Diatoms increase by 22% in the SAZ (north of the SAF) while there are no significant changes in coccolithophores abundance. As a result of the changes in these two plankton species, EP increases by 1.3% in the SAZ and decreases by 14% in the AZ (Figure 3b). The total Southern Ocean (south of 30°S) EP and NPP decrease by 2% and 7.5%, respectively at 70 ka.

## 3.2 Impact of changes in iron dust flux on atmospheric $CO_2$

Changing the iron flux masks from PI to glacial at 70 ka BP leads to a 3.8 to 8.3 ppm drop in atmospheric $CO_2$ concentration if we assume that the mean iron solubility remains unchanged (Table 2). Interestingly, for solubilities of 3% and higher, the drawdown is nearly constant, regardless of the glacial dust flux mask and regardless of the solubility ($7.3 \pm 1$ ppm).

However, the solubility of iron was likely higher during cold than warm periods (see section 2.2). We will therefore discuss experiments that switch from a PI iron mask with 1% solubility to glacial iron masks with higher solubilities from hereon. These experiments show a $\sim$9 to 19 ppm drop in atmospheric $CO_2$ concentration (Figure 4a and b, Table 3). These changes in atmospheric $CO_2$ impact the physical conditions of the ocean only slightly in our model. As such, the global overturning circulation, mean SST and sea-ice extent are similar in most of the sensitivity experiments described below.

As the global and Southern Ocean iron input increase, the atmospheric $CO_2$ concentration decreases, but not linearly, as the efficiency of the iron fertilisation weakens with the increasing iron availability (Figure 4a,b). The patterns of aeolian iron deposition in the Southern Ocean are different for the two masks used here. Both reconstructions show an increase in the South Atlantic sector compared to PI fluxes, but they differ over the Pacific and Indian sectors, including large regions where dust deposition decreases compared to PI (Figure 2). It is therefore interesting that our experiments with similar total iron input into the Southern Ocean but different regional patterns (e.g. glacfe20% and lambfe20%, Figure 4b, pink star and black circle) display similar efficiency in drawing down $CO_2$ regardless of the dust mask used. The glacfe mask is however slightly more efficient in drawing down atmospheric $CO_2$. The iron flux into the South Atlantic sector is higher in the glacfe mask compared to the lambfe mask, increasing iron concentrations near one of our major convection sites in the Weddell Sea. Bottom water

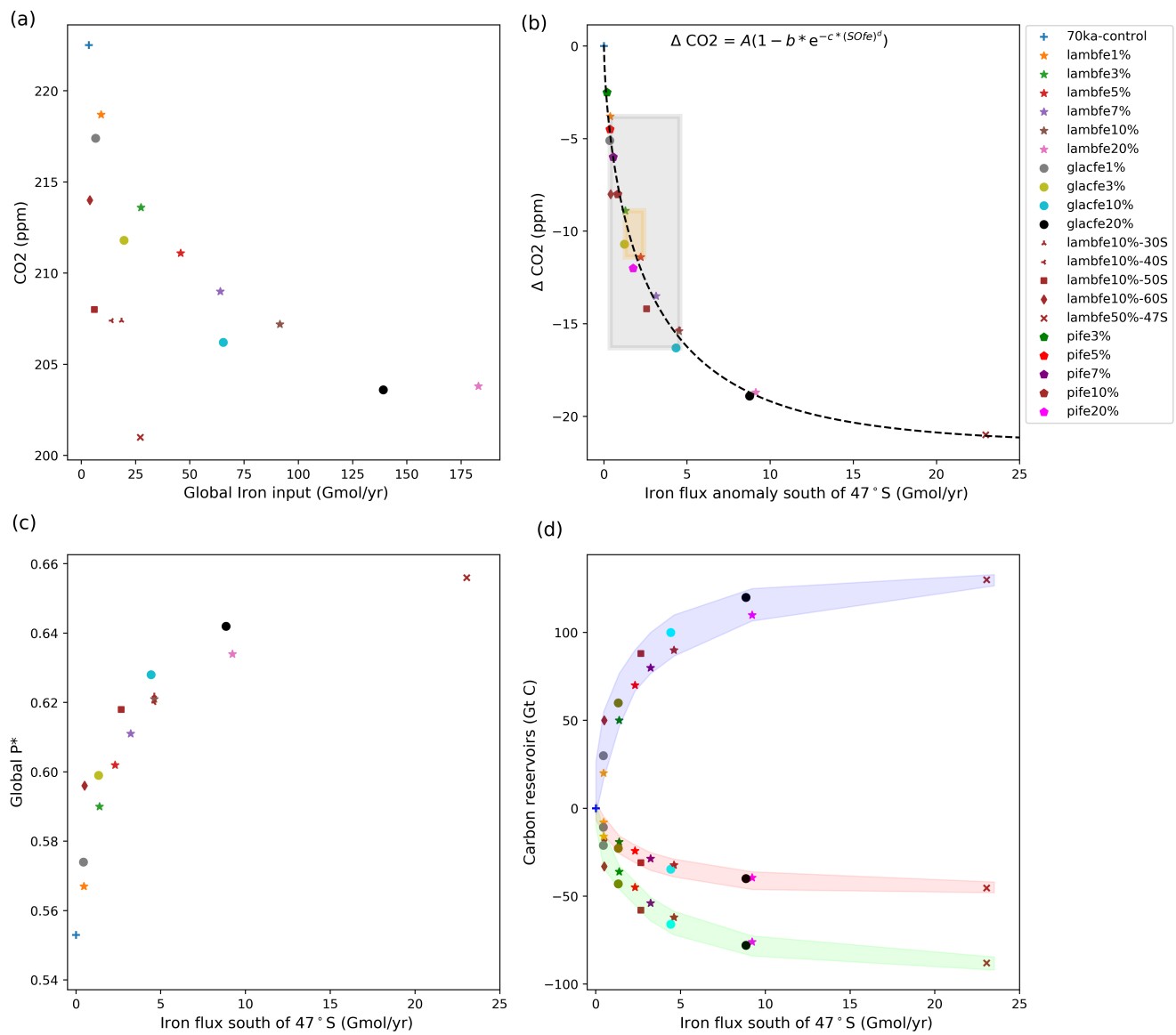

**Figure 4.** (a) Equilibrated atmospheric $CO_2$ concentration (ppm) as a function of the globally integrated aeolian iron flux into the ocean (Gmolyr$^{-1}$). (b) Atmospheric $\Delta CO_2$ concentration (ppm) as a function of changes in aeolian iron flux into the Southern Ocean south of 47°S. The grey shading represents the range of likely glacial iron solubilities (1-10%) and the associated change in $CO_2$ concentration (-4 to -16 ppm), while the orange shading represents our best estimate of change in $CO_2$ (-9 to -11 ppm) for a solubility of 1% during warm periods and 3-5% during colder periods. Note that lambfe10%, lambfe10%-30S and lambfe10%-40S are overlapping. The black curve represents the best fit and suggests a maximum $CO_2$ draw-down of ∼21 ppm due to Southern Ocean iron fertilisation. (c) Global P* as a function of aeolian iron flux into the Southern Ocean south of 47°S and (d) Globally integrated carbon reservoirs (GtC) as a function of aeolian iron flux into the Southern Ocean south of 47°S. The shadings represent ocean carbon (blue), atmospheric carbon (red) and terrestrial carbon (green).

**Table 3.** $\Delta CO_2$ for the sensitivity experiments compared to 70 ka-control, globally averaged P* values, as well as NPP and EP values integrated over different regions of the Southern Ocean. Percentage changes from 70ka-control experiment are provided in brackets

| Experiment | $\Delta CO_2$ (ppm) | Global P* | SO NPP $(30°S:90°S,$ Pg $Cyr^{-1}$) | SO NPP north of $47°S$ (Pg $Cyr^{-1}$) | SO NPP south of $47°S$ (Pg $Cyr^{-1}$) | SO EP $(30°S:90°S,$ Pg $Cyr^{-1}$) | SO EP north of $47°S$ (Pg C $yr^{-1}$) | SO EP south of $47°S$ (Pg C $yr^{-1}$) |
|---|---|---|---|---|---|---|---|---|
| PI-control (pife1%) | - | 0.515 | 14.63 | 7.62 | 7 | 3.84 | 1.58 | 2.267 |
| 70ka-control (pife1%) | 0 | 0.553 | 13.52 | 5.38 | 8.15 | 3.76 | 1.13 | 2.63 |
| lambfe1% | -3.8 | 0.566 | 13.58 (+0.4%) | 4.55 (-15%) | 9.04 (+11%) | 3.93 (+4.5%) | 0.99 (-12%) | 2.93 (+11.6%) |
| lambfe3% | -8.9 | 0.590 | 13.43 (-0.6%) | 3.77 (-30%) | 9.68 (+18.7%) | 4.08 (+8.6%) | 0.84 (-25%) | 3.23 (+23%) |
| lambfe5% | -11.4 | 0.602 | 13.36 (-1.1%) | 3.47 (-35%) | 9.89 (+21%) | 4.14 (+10.3%) | 0.79 (-30%) | 3.36 (+30%) |
| lambfe7% | -13.5 | 0.611 | 13.31 (-1.5%) | 3.31 (-38%) | 10.02 (+23%) | 4.19 (+11.4%) | 0.75 (-33%) | 3.44 (+30.7%) |
| lambfe10% | -15.3 | 0.621 | 13.27 (-1.8%) | 3.16 (-41%) | 10.12 (+24%) | 4.22 (+12.4%) | 0.72 (-36%) | 3.5 (+33%) |
| lambfe20% | -18.7 | 0.634 | 13.35 (-1.2%) | 3.01 (-44%) | 10.35 (+27%) | 4.33 (+15%) | 0.68 (-39%) | 3.64 (+38.5%) |
| glacfe1% | -5.1 | 0.574 | 13.39 (-0.9%) | 4.43 (-17.6%) | 8.97 (+10%) | 3.92 (+4.3%) | 0.97 (-13.7%) | 2.94 (+12%) |
| glacfe3% | -10.7 | 0.599 | 13.21 (-2.2%) | 3.60 (-33%) | 9.62 (+18%) | 4.07 (+8.2%) | 0.81 (-28%) | 3.25 (+23.8%) |
| glacfe10% | -16.3 | 0.628 | 13.14 (-2.8%) | 3.12 (-42%) | 10.03 (+23%) | 4.20 (+11.7%) | 0.71 (-37%) | 3.5 (+32.8%) |
| glacfe20% | -18.9 | 0.642 | 13.09 (-3.1%) | 2.98 (-45%) | 10.12 (+24%) | 4.25 (+13%) | 0.68 (-40%) | 3.57 (+36%) |

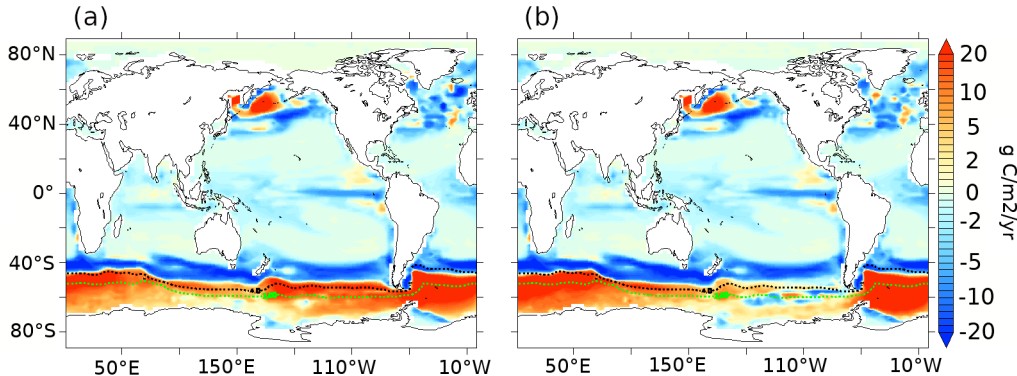

**Figure 5.** Export production anomalies (gC $m^{-2}$ $yr^{-1}$) at 177.5m depth for (a) lambfe3% and (b) glacfe3% compared to 70ka-control (pife1%). Overlaid dashed contours represents modern SAF in black and APF in green based on Sokolov and Rintoul (2009)

formation in this region leads to greater mixing with deeper layers, replenishing surface nutrient concentrations and resulting in higher export production (Marinov et al., 2006). Global P* is ∼1.5% higher in our glacfe experiments than in our lambfe experiments resulting in a slightly larger $CO_2$ drawdown.

The sensitivity experiments provide additional information on the role of the Southern Ocean in our simulations. We find that experiment lambfe10% is as efficient in drawing down $CO_2$ as the experiments where iron fluxes were only enhanced south of 30°S or 40°S (lambfe10%-30S, lambfe10%-40S) (Figure 4a,b)). This indicates that in our experiments iron fertilization in the Southern Ocean south of 40°S is mainly responsible for the atmospheric $CO_2$ draw-down while fertilization elsewhere plays a negligible role. On the other hand, a 14 ppm and 8.5 ppm $CO_2$ decrease is simulated in lambfe10%-50S and lambfe10%-60S, respectively, compared to 15.3 ppm in lambfe10%.

Our experiments show that the ecosystem response north and south of ∼47°S (which is roughly the modern SAF) are of opposite sign (Table 3). While an increase in iron availability leads to an increase in NPP, diatom and coccolithophore

abundance, and consequently EP south of the SAF (Figure 5 and 6), the regions north of the SAF show a significant decline in all of these parameters. In our model, the EP increase occurs south of the SAF, where the surface nutrient concentrations are high (Figure A5a). Enhanced aeolian iron input leads to a more efficient use of nutrients in that region and results in a reduction in northward transport of nutrients north of the polar front (Figure A6) which in turn leads to a decrease in EP in the SAZ.

The relationship between atmospheric $CO_2$ and the iron flux south of $47°S$ follows (Figure 4b):

$$\Delta CO_2 = a(1 - b \times e^{-c \times (SOfe)^d}) \tag{5}$$

where $\Delta CO_2$ is the atmospheric $CO_2$ anomaly in ppm compared to 70ka-control, SOfe denotes the total iron input into the Southern Ocean south of $47°S$ in $Gmolyr^{-1}$, a= -21.51 ppm, b= 1.0, c= 0.487 $(Gmolyr^{-1})^{-d}$ and d= 0.66.

We also note that the $CO_2$ draw-down capacity levels out for high iron forcing. Both lambfe and glacfe masks with 20%
iron solubility lead to a $\sim$19 ppm $CO_2$ decrease. In our extreme experiment lambfe50%-47S, with an iron input south of $47°S$ almost 2.5 times higher than in lambfe20% and glacfe20%, only an additional 2 ppm $CO_2$ draw-down is simulated (Figure 4b). Our results therefore suggest that iron fertilization could have led to a maximum $CO_2$ decrease of $\sim$19-21 ppm at 70 ka.

The enhanced iron flux leads to an increase in $P_{reg}$ in the Southern Ocean, which is subsequently transported northward at depth by AABW. As a result, the globally averaged $P^*$, and therefore the overall efficiency of the soft tissue pump, increases
with Southern Ocean aeolian iron input (Figure 4c). As ocean circulation and ocean temperatures change only slightly in our experiments, the relationship between $P^*$ and the resulting $CO_2$ draw-down is almost linear (Figure A7) (Ito and Follows, 2005).

However, as the Southern Ocean iron flux increases, the overall iron fertilisation efficiency reduces. While export production increases south of $47°S$, thus using nutrients more efficiently (Figure A8c) and reducing the nutrient advection north of the
SAF (Figure A8b), nitrate limitation increases in the SAZ (Figure A9b,d and Figure A10a,b), leading to a decrease in export production. For example, in experiment lambfe20%, a 67% decrease in nitrate concentration is simulated between $40°S$ and $47°S$ (Figure A8b). At the same time, silicate limitation decreases in the SAZ (Figure A9e and Figure A10c) due to a southward shift of both coccolithophores and diatoms, and a decrease in diatoms between $40°S$ and $55°S$ (Figure A11). This further enhances silica availability in this region, consequently leading to a decrease in silicate limitation. Therefore, the total biological
pump efficiency, represented here by changes in $P^*$ (Figure 4c) and Southern Ocean EP to NPP ratio (Figure A8a), saturate at high iron values due to nitrate limitation north of $50°S$ in the Southern Ocean. The global $P^*$ in lambfe20% and in lambfe50%-47S (Figure 4c) equal 0.63 and 0.65 respectively, suggesting a maximum efficiency of 65% in our experimental set-up.

The oceanic carbon reservoir in our sensitivity experiments increases between 20 and 130 GtC depending on the iron flux scenario (Figure 4d, blue shade). The simulated decrease in atmospheric $CO_2$, equivalent to 8-45 GtC (Figure 4d, red shade)
leads to a decrease in surface air temperatures, as well as regional changes in precipitation and soil moisture. In addition, the lower atmospheric $CO_2$ concentration also reduces photosynthesis and consequently litter fall. The direct and indirect effects of a lower atmospheric $CO_2$ concentration result in a terrestrial carbon decrease of 16 to 88 GtC (Figure 4d, green shade), out of which 8 to 45 GtC decrease is from terrestrial vegetation while 8 to 43 GtC reduction is from soil carbon.

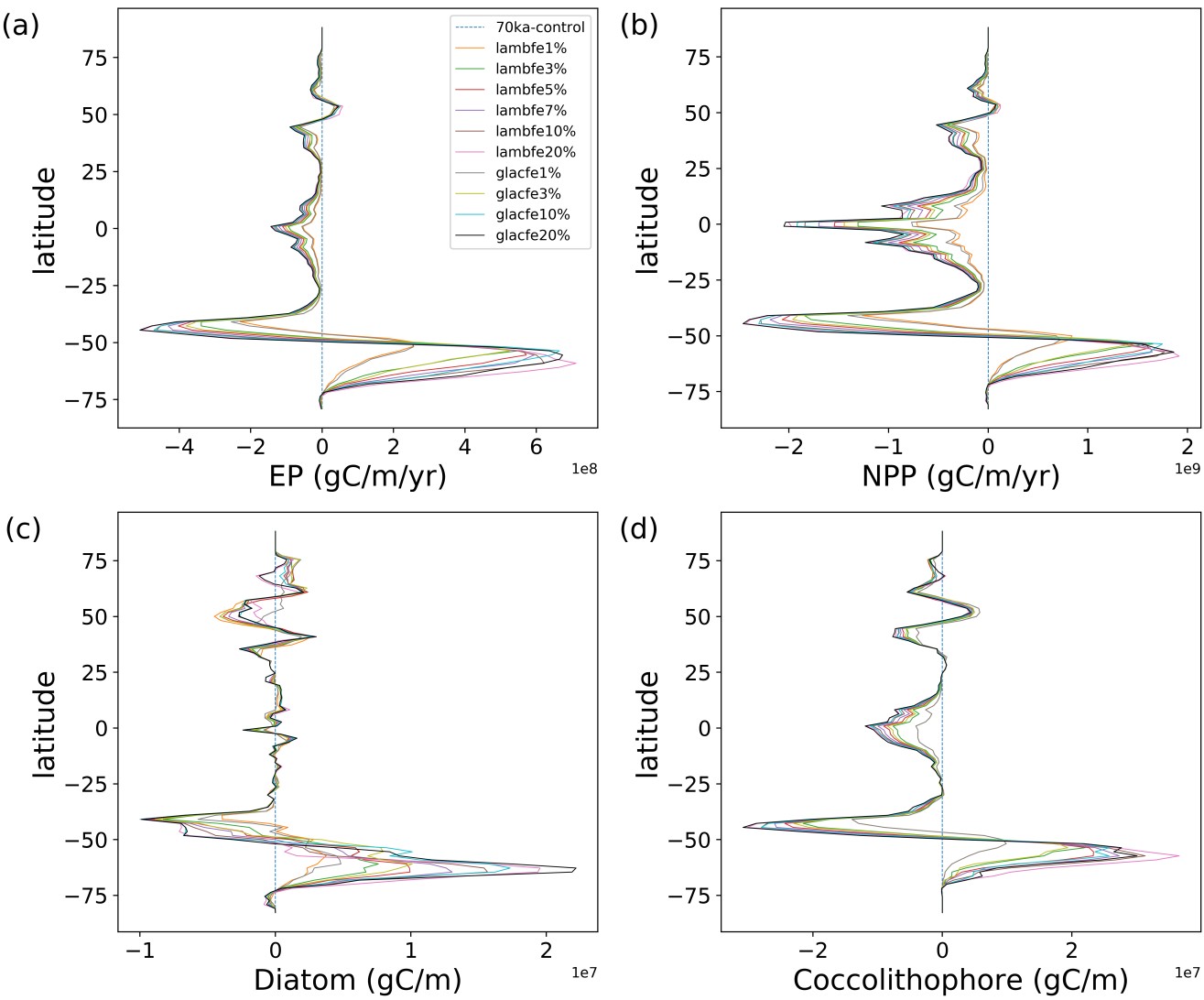

**Figure 6.** Zonally integrated anomalies of (a) EP (gC m$^{-1}$yr$^{-1}$) at 177.5m, (b) NPP (gC m$^{-1}$yr$^{-1}$), (c) Diatoms (gC m$^{-1}$) and (d) Coccolithophores (gC m$^{-1}$), compared to 70ka-control. Please note the multiplication factor allocated to each subpanel.

### 3.3 Impact of aeolian iron input on the distribution of oceanic carbon

As mentioned in the previous section, our sensitivity experiments do not show significant changes in the physical ocean conditions. However, enhanced iron input significantly impacts marine ecosystems. To understand the resulting ocean carbon changes, we start by describing changes in the simulated ecosystems for one of our sensitivity experiments, lambfe3%, compared to 70ka-control. We choose lambfe3% (Figure A2) because earlier research shows that the lambfe Southern Ocean dust mask provides a better fit with available glacial dust flux proxy records than the glacfe mask (Saini et al., 2021). Furthermore,

a solubility factor of 3% corresponds to a likely estimate of iron solubility at 70 ka (see Methods) and has also been used in previously studies (Tagliabue et al., 2014; Lambert et al., 2015; Muglia et al., 2017; Yamamoto et al., 2019). All lambfe experiments with different solubility factors show similar anomaly patterns, only the magnitude of the changes varies as a function of solubility (Figure A11).

A $\sim$9 ppm drop in atmospheric $CO_2$ is simulated in experiment lambfe3% with a global increase in EP by 1.3%, and a global

decrease in NPP by 6.7% (Table 3 and Table A1). Iron increase in the lambfe3% experiment leads to a 37% increase in diatoms and an 88% increase in coccolithophores in the AZ (Figure A4c,d). The simulated increased EP in the AZ (Figure 5a) leads to greater nutrient utilisation south of the APF (Figure A8c). On the contrary, because of lower nutrient availability, both diatoms and coccolithophores abundances decrease in the SAZ by 46% and 31% respectively (Figure A4c,d). Consequently, in the lambfe3% experiment, the EP increases by 98% in the AZ while it decreases by 17% in the SAZ compared to the 70ka-control

simulation (Figure 5a, 6a).

The overall Southern Ocean EP south of 30°S increases by $\sim$9% in the lambfe3% experiment, the NPP changes become insignificant (Table 3), and the total biological pump efficiency increases by 6.7% (Table 3). In the North Pacific (150°E:220°E;47°N:56°N), EP increases by 13%, while NPP increases by 7% due to a 82% increase in coccolithophores and 58% decrease in diatoms. Both NPP and EP decrease by $\sim$10% in the North Atlantic (60°W:0°E;47°N:56°N) where we

see an overall decrease in coccolithophores by 22% and an increase in diatoms by 16%.

The increased carbon export from the surface into the deep ocean increases DIC at all depths south of 47°S. The DIC rich bottom water is advected northward and leads to a stronger vertical DIC gradient in the ocean (Figure 7a and b). While DIC increases by 25 mmol m$^{-3}$ in the deep Southern Ocean (south of 30°S, below 3000m), it does not increase uniformly across sectors: it increases by 22 mmol m$^{-3}$ in the deep Indo-Pacific sector (below 3000m, Figure 7a), and by 29 mmol m$^{-3}$ in the

deep Atlantic sector (below 3000m, Figure 7b). The positive DIC anomalies north of 30°S are associated with AABW and its northward spread into the Atlantic and the Indo-Pacific basins, therefore leading to a 10 mmol m$^{-3}$ DIC increase in the deep Indo-Pacific and a 15 mmol m$^{-3}$ increase in the deep Atlantic sector.

In contrast, lower DIC is simulated north of 47°S in the upper Pacific ocean (above 2000m), and within the North Atlantic Deep Water (NADW) and Antarctic Intermediate Water (AAIW) pathways in the Atlantic above 3000m. Simulated changes

in deep ocean oxygen concentrations (not shown) are opposite to the DIC changes indicating an increase in remineralised carbon in the deep ocean (Figure 8a,b). An increase in mixed layer alkalinity (Figure 7c and d) is simulated, partially due to the decrease in coccolithophores between 47°S-50°N (Figure 6d and A4d). As a result of these changes in surface $CaCO_3$

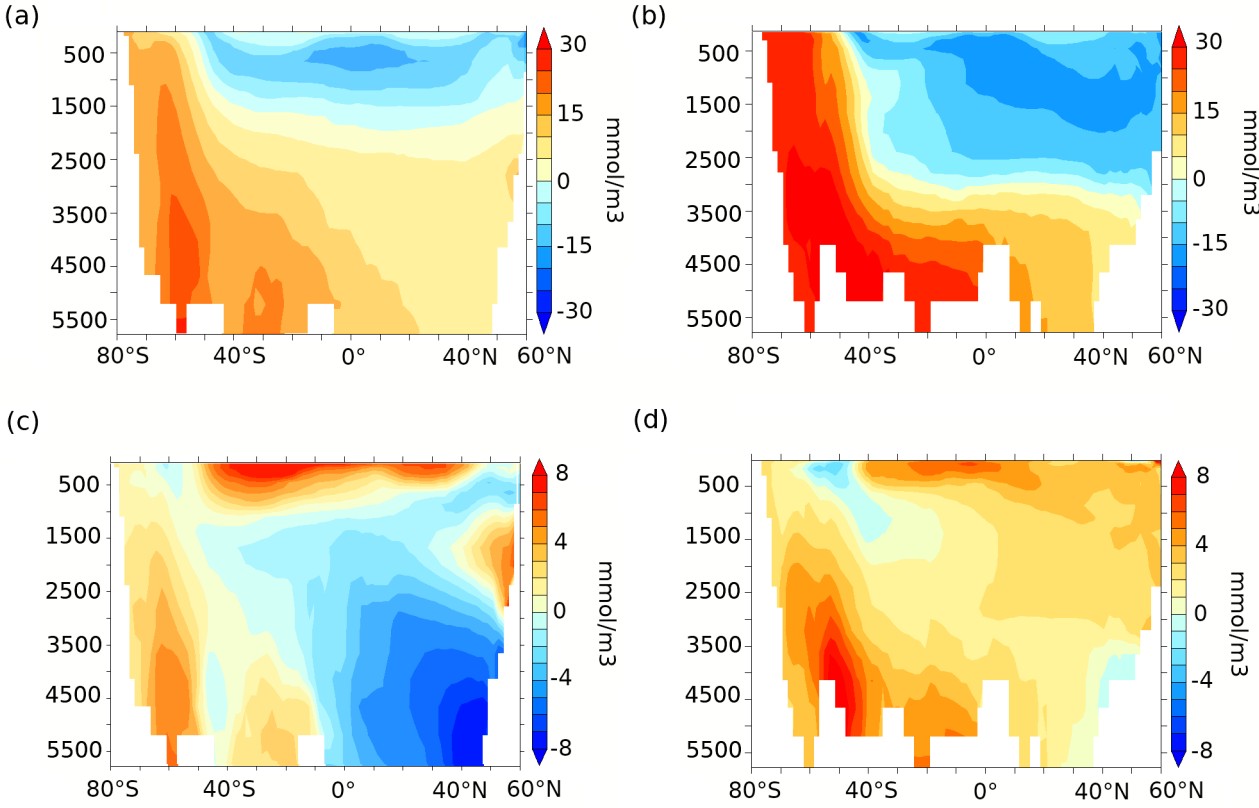

**Figure 7.** Zonally averaged DIC anomalies (mmol m$^{-3}$) over (a) the Indo-Pacific and (b) Atlantic, and zonally averaged alkalinity anomalies (mmol m$^{-3}$) over (c) the Indo-Pacific and (d) the Atlantic for lambfe3% compared to 70ka-control.

formation and subsequent changes in the subduction and sinking of CaCO$_3$ into deeper layers, reduced CaCO$_3$ dissolution at depth leads to lower alkalinity in the North Pacific, while higher CaCO$_3$ dissolution in the Southern Ocean and in the deep Atlantic leads to an alkalinity increase there (Figure 7c, d and Figure 8c, d).

To quantify the processes leading to changes in oceanic DIC, we decompose the changes in DIC into their remineralized, carbonate and preformed contributions (see Methods). In our 70ka-control simulation, the remineralization process leads to the generation of 131 mmol m$^{-3}$ C$_{reg}$ in the deep ($\geq$3000m depth) Southern Ocean, which increases by 22.5 mmol m$^{-3}$ in lambfe3% due to higher iron influx. Changes in the carbonate pump contribute marginally to the deep Southern Ocean DIC showing an increase of 3.16 mmol m$^{-3}$ C$_{CaCO_3}$, whereas the preformed DIC contribution (C$_{pref}$) decreases by 0.66 mmol m$^{-3}$, compared to 70ka-control. North of 30°S, C$_{reg}$ in lambfe3% increases by 14.7 mmol m$^{-3}$ in the deep Indo-Pacific basin while both C$_{CaCO_3}$ and C$_{pref}$ decrease by 2 mmol m$^{-3}$ and 2.6 mmol m$^{-3}$ respectively (Figure 8a, c, e). The deep Atlantic

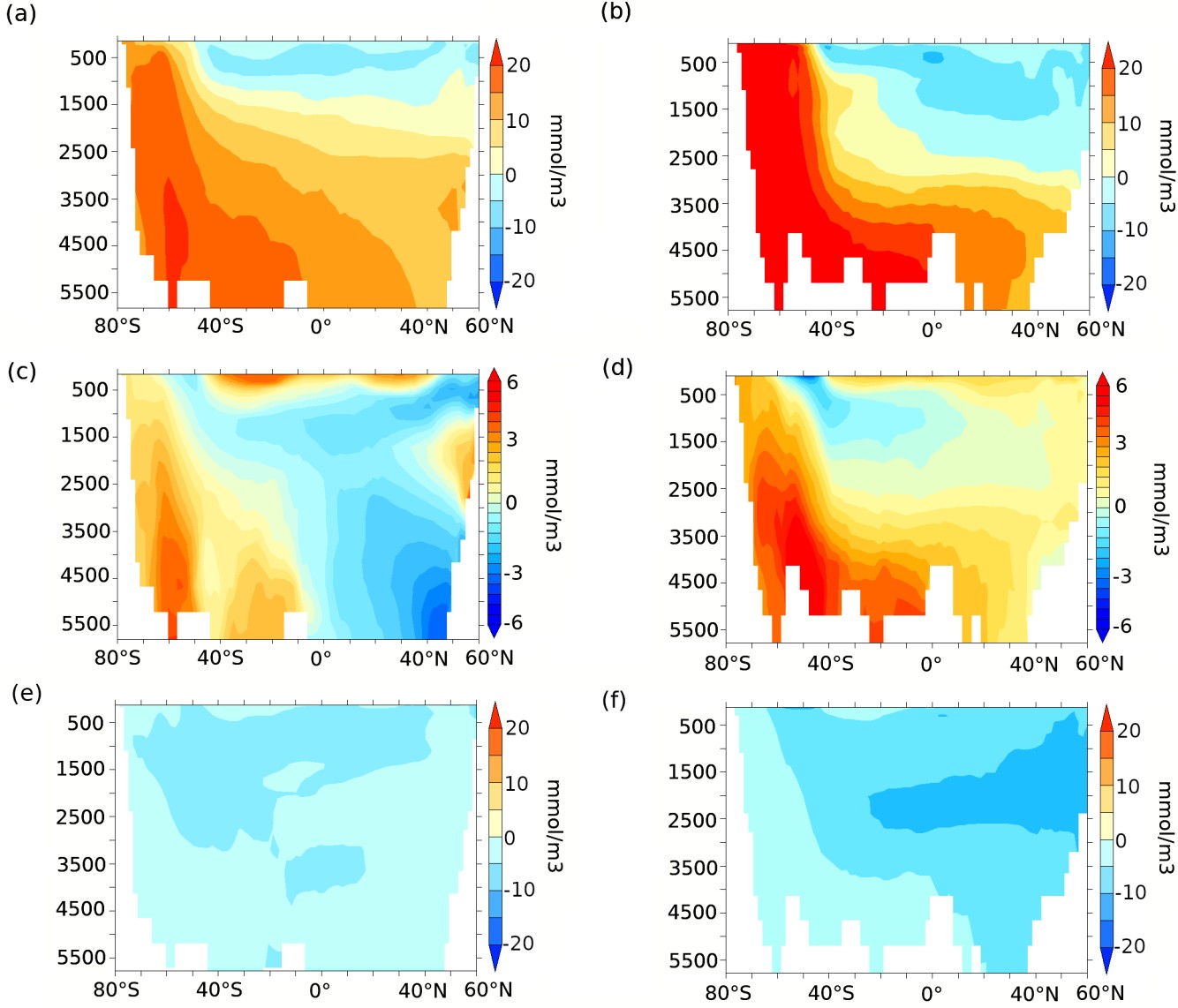

**Figure 8.** Zonally averaged (a, b) remineralized carbon (mmol m$^{-3}$), (c, d) carbon due to calcite dissolution (mmol m$^{-3}$) and (e, f) preformed carbon (mmol m$^{-3}$) anomalies for lambfe3% compared to 70ka-control in the Indo-Pacific (left panels) and the Atlantic (right panels). Please note that the color scale is different for subpanels c and d compared to other subpanels.

ocean shows an increase in C$_{reg}$ by 16.3 mmol m$^{-3}$ and in C$_{CaCO_3}$ by 2 mmol m$^{-3}$ in lambfe3% experiment, while C$_{pref}$ decreases by 3.2 mmol m$^{-3}$ (Figure 8b, d, f).

## 4 Discussion

The colder climate at 70 ka BP leads to an overall decrease in NPP and EP compared to PI in our simulations (9.7% and 3%, respectively). The changes in NPP and EP are spatially heterogeneous and sensitive to changes in the length of the growing season, sea ice cover, and nutrient availability and are thus associated with shifts in plankton distribution. In the Southern Ocean, the ~7.5% reduction in NPP and 2% reduction in EP might be due to a shorter growing season for diatoms and coccolithophores (Saini et al., 2021), associated with a larger annual sea ice extent and lower SSTs at 70 ka BP. Both NPP and EP also decrease at low and mid latitudes.

Alkenone flux (Lamy et al., 2014; Martínez-García et al., 2014), as well as opal and organic carbon fluxes (Thöle et al., 2019; Amsler et al., 2022) reconstructed from subantarctic sediment cores suggest that EP was higher in the SAZ during MIS4 than present day. In addition, bottom water oxygenation records indicate a deep ocean oxygenation decrease during MIS4, which might suggest increased respired carbon storage in the deep ocean (Amsler et al., 2022) but could also reflect a change in ocean dynamics and water residence time. It has been suggested that the subantarctic EP increase during MIS4 was due to higher iron fluxes into the Southern Ocean. Higher dust deposition in the South Pacific and the South Atlantic at 70 ka is consistent with available paleo-proxy records (Martínez-García et al., 2011; Lambert et al., 2012; Lamy et al., 2014; Thöle et al., 2019). At the same time, palaeoceanographic records from the AZ suggest a decrease in EP during MIS4 (Anderson et al., 2009; Jaccard et al., 2013; Studer et al., 2015; Thöle et al., 2019; Amsler et al., 2022; Weber et al., 2022).

In our 70 ka simulations with enhanced iron input, EP increases in the AZ and polar frontal zone (between SAF and APF) and decreases in the SAZ, in contrast with most paleo-proxy records (Figure A5b). Existing $\delta^{15}$N records suggest higher nutrient consumption at the MIS4/MIS5 transition in the AZ (Studer et al., 2015; Ai et al., 2020), a region where the supply of macronutrients to the surface is determined by ocean upwelling and mixing (Lefèvre and Watson, 1999; Tagliabue et al., 2017). Studer et al. (2015) argues that the relative increase in nutrient utilisation in the AZ during MIS4 is due to the general decrease in the nitrate supply, resulting from greater isolation of the deep ocean during glacial periods (Francois et al., 1997; Jaccard et al., 2013; Sigman et al., 2021), rather than iron fertilisation.

This is in contradiction to our results, where we find a deepening of the winter mixed layer in the 70 ka simulations compared to the PI-control, which leads to a slight increase in macronutrients in the AZ. Higher iron supply within this diatom-dominated seasonal sea ice zone then leads to a greater utilisation of available nutrients, in agreement with previous studies (Abelmann et al., 2006, 2015), resulting in greater EP within the AZ. With higher nutrient utilisation in the AZ, the nutrient advection into the SAZ is reduced, leading to a decrease in EP in the SAZ. The disagreement between the modelling results and paleo-proxy records thus suggests that the AZ is not stratified enough in our 70ka simulations. This could be due to a weakening and/or more equatorward position of the SH westerly winds during glacial times (e.g., Toggweiler et al. (2006); Gray et al. (2023)), which are not included in our simulations as present-day surface winds are prescribed.

However, our simulated EP increase in the AZ during MIS4 compared to PI, and thus the increase in regenerated carbon storage in the deep ocean, aligns with some proxy records suggesting a decrease in deep ocean oxygenation in the AZ (Jaccard et al., 2016; Amsler et al., 2022). Our EP increase is also consistent with the increase in opal flux north of the APF (Amsler

et al., 2022) at the MIS4 onset (Figure A5b). In summary, while our EP responses in the SAZ and AZ are inconsistent with most of the paleo records, the strengthening of the biological pump leading to greater storage of deep ocean organic carbon is consistent with palaeoceanographic evidence. Since proxy data for dust flux and EP is limited to only a few marine sites and ice cores, additional paleo-proxy records covering a larger area in the Southern Ocean during MIS4 are needed to better quantify the impact of iron fertilization during the glaciation.

Our results suggest that enhanced iron input south of the Antarctic polar front, i.e. south of ∼47°S (Dong et al., 2006; Sokolov and Rintoul, 2009; Giglio and Johnson, 2016), is responsible for 100% of the simulated atmospheric $CO_2$ drawdown. This is in agreement with the hypothesis put forward by Marinov et al. (2006), which suggests that changes in the efficiency of the biological pump in the Antarctic zone play a dominant role in $CO_2$ drawdown. However, our results differ from Lambert et al. (2021), who find that ∼30% of the simulated $CO_2$ decrease during the last glacial termination is due to enhanced aeolian iron input into the North Pacific in their simulations. While they use the same LGM dust flux, their iron input in the ocean differs from our study. Lambert et al. (2021) assume 3.5 wt% iron fraction in the aeolian dust flux, whereas we use a global map of percentage distribution of iron in dust (see Methods). More importantly, their solubility factor is not constant, but varies non linearly with the magnitude of the dust flux.

While the two dust masks used here suggest an increase in aeolian iron deposition in the Atlantic sector of the Southern Ocean during glacial times compared to PI, the iron input in glacfe is lower in the South Pacific compared to pife, whereas lambfe is higher in the Pacific sector. Despite these differences in spatial patterns, the two iron masks (glacfe and lambfe) with the same iron solubilities lead to similar decreases in atmospheric $CO_2$ in our simulations. This indicates that changes in atmospheric $CO_2$ are more dependent on changes in solubility, than on regional differences in aeolian iron fluxes in the Southern Ocean. The experiments using the glacfe mask are slightly more efficient in drawing down $CO_2$. One of the major Antarctic Bottom Water formation sites is located in the Weddell Sea in our simulations, making the South Atlantic sector a more efficient region for carbon sequestration. Higher iron dust input in the Weddell Sea in the glacfe experiments thus leads to greater $CO_2$ drawdown compared to the lambfe experiments.

With no significant ocean circulation changes, changes in atmospheric $CO_2$ are a linear function of the overall efficiency of the biological pump ($P^*$, Figure A7). In agreement with earlier studies (Matsumoto et al., 2002; Tagliabue et al., 2014; Lambert et al., 2015; Yamamoto et al., 2019), we find that the efficiency of the soft tissue pump increases south of 47°S, while it decreases in the lower latitudes due to reduced northward advection of surface nutrients. The increase in the vertical nutrient gradient leads to the eventual saturation of the ocean carbon uptake capacity.

We simulate a maximum of 16-18% increase in global $P^*$ in our experiments, leading to a maximum drop of 19 to 21 ppm in $CO_2$. The timescale of this $CO_2$ draw-down in our idealised simulations, i.e. without transient changes in dust or transient changes in ice sheet volume, is ∼5000 years, which is consistent with the observed timescale of $CO_2$ transitions during MIS4 (Bereiter et al., 2015). If we define the plausible range of the large-scale average iron solubility in the ocean to be 1 to 10% at 70 ka BP, the corresponding range of atmospheric $CO_2$ draw-down is ∼4 to 16 ppm. More likely, large-scale solubility in the Southern Ocean at 70 ka ranged between 3 and 5%, which leads in our simulations to a $CO_2$ decline of 9 to 11 ppm (Figure 4b).

We find that the biological response to changes in iron fertilization not only depends on the iron solubility during glacial periods but also on the iron solubility during warm periods. Our results are based on the assumption that the global average iron solubility during warm periods equals ∼1%. At higher initial values, the total potential draw-down of $CO_2$ would be smaller. For example, for an assumed solubility during warm periods closer to ∼3%, we simulate a range of $CO_2$ changes between 6.4 ppm (no change in solubility and glacial fluxes based on Lambert et al. (2015)) and 16.4 ppm (change to 20% solubility and glacial fluxes based on Ohgaito et al. (2018)). This range reduces to 6.9 to 14.4 ppm if the initial solubility was 5% and to 6.7 to 6.9 ppm if the initial solubility was 20%.

Previous modeling studies obtained a 2 ppm drop in $CO_2$ using 2% iron solubility in the Southern Ocean under PI boundary conditions (Tagliabue et al., 2014), while under LGM boundary conditions, using solubility factors between 1 and 2%, a 2-28 ppm drop in $CO_2$ was simulated by a hierarchy of models (Bopp et al., 2003; Tagliabue et al., 2009b; Lambert et al., 2015; Muglia et al., 2017; Yamamoto et al., 2019). Using iron solubility factors of 3% and 10% under LGM conditions,Yamamoto et al. (2019) simulated a 15.6 ppm to 20 ppm $CO_2$ reduction, while a solubility of 7% with varying dust masks led to a decrease in $CO_2$ by 12-29 ppm in Lambert et al. (2021). A 12 ppm decrease at 70 ka was simulated in a transient simulation performed with a model of intermediate complexity using 1% solubility (Menviel et al., 2012). Our results are consistent but in the lower range of previous modelling estimates. Although our lowest estimate of a 4 ppm $CO_2$ reduction is in agreement with the studies mentioned above, the upper limit of 16 ppm is lower than some previous estimates (e.g. Yamamoto et al. (2019), Lambert et al. (2021)). This might be due to the comparatively larger simulated decrease in EP north of 47°S and in the equatorial Pacific in our study compared to previous ones (Yamamoto et al., 2019).

None of the previous modelling studies on iron fertilisation simulate coccolithophores and diatoms' abundances prognostically. By including four distinct classes of plankton in our model, we highlight the competitive dynamics between different major phytoplankton functional types for light and nutrient availability. Coccolithophores contribute to the total carbon export mainly in the polar frontal zone, while diatoms' contribution is in the Antarctic zone. As previously mentioned, carbon export close to convection sites in the Southern Ocean can be more efficient in reducing atmospheric $CO_2$. Furthermore, while both diatom and coccolithophores contribute to $CO_2$ uptake in the ocean through photosynthesis, coccolithophores produce $CaCO_3$ rich platelets, which reduce surface ocean alkalinity, thus reducing the $CO_2$ uptake efficiency. The inclusion of this competition should be taken into account when investigating the impact of ecosystem changes on the global carbon cycle.

Our estimate of a 16 ppm $CO_2$ decrease with 10% iron solubility corresponds to half of the ∼32 ppm $CO_2$ drop estimated for the MIS4 transition (Bereiter et al., 2015). This suggests that enhanced aeolian iron input at the MIS4 transition could have played a significant role in the $CO_2$ draw-down (Martínez-García et al., 2014; Kohfeld and Chase, 2017), but other processes must also have contributed to the $CO_2$ decline. For example, an AMOC weakening (Piotrowski et al., 2005; Wilson et al., 2015; O'Neill et al., 2021) could have enhanced ocean stratification, contributing 15-30 ppm to the $CO_2$ decrease during MIS4 (Thornalley et al., 2013; Yu et al., 2016; Menviel et al., 2017). An increase in ocean stratification at the MIS4 transition due to changes in AABW properties (Adkins, 2013) could also have increased the deep ocean carbon content (Menviel et al., 2017), thus contributing to the atmospheric $CO_2$ decline. Finally, both changes in oceanic circulation and a more efficient biological

pump could have enhanced sediment calcite dissolution, thus increasing ocean alkalinity and enhancing the $CO_2$ draw-down (Boyle, 1988; Archer and Maier-Reimer, 1994; Yu et al., 2016; Kobayashi and Oka, 2018).

## 5 Conclusions

We use an Earth system model of intermediate complexity incorporating a newly developed ecosystem model to better constrain the oceanic carbon uptake due to enhanced aeolian iron fluxes at the MIS4 transition. Based on a series of 70 ka sensitivity experiments forced with two different aeolian iron flux masks and different iron solubility factors, we find that enhanced iron input south of 47°S could have led to a 4 to 16 ppm $CO_2$ decline at 70 ka BP, with a most likely range of 9 to 11 ppm. Our results suggest that the ocean's capacity to take up carbon decreases with increasing iron input because enhanced nutrient

utilisation in the Antarctic zone is compensated by decreased nutrient availability at mid and low latitudes. Enhanced aeolian iron input at the MIS4 transition could thus explain up to 50% of the observed $CO_2$ drop at that time.

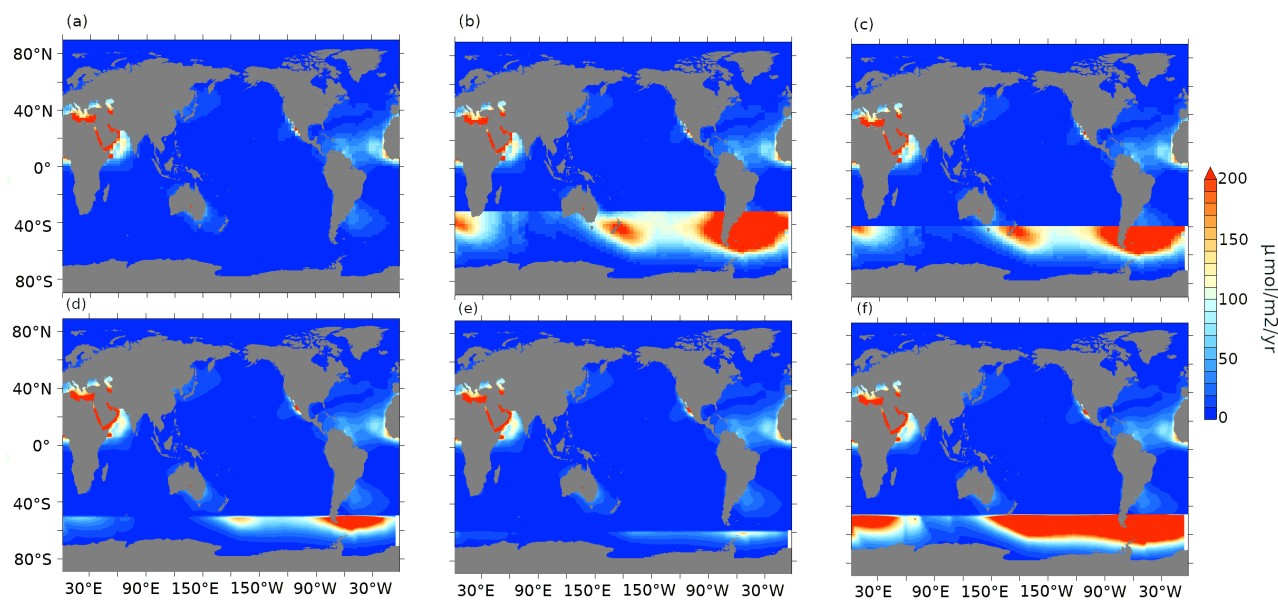

**Figure A1.** Aeolian iron dust flux ($\mu$mol m$^{-2}$ yr$^{-1}$) for (a) pife1%, (b) lambfe10%-30S (iron flux equals lambfe10% south of 30°S and pife1% elsewhere), (c) lambfe10%-40S (same as lambfe10%-30S but for 40°S), (d) lambfe10%-50S, (e) lambfe10%-60S, (f) lambfe50%-47S (iron flux equals to lambfe50% south of 47°S and pife1% elsewhere).

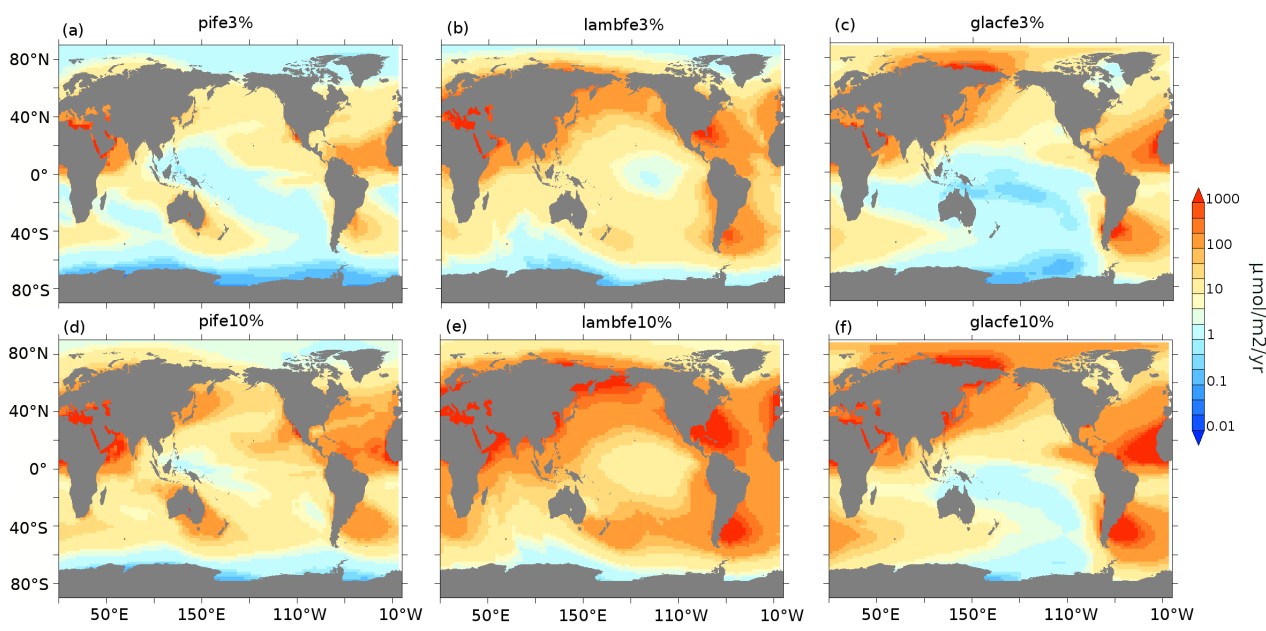

**Figure A2.** Aeolian iron dust fluxes ($\mu$mol m$^{-2}$ yr$^{-1}$) for pife (a,d), lambfe (b,e) and glacfe (c,f) masks with 3% (top row) and 10% (bottom row) solubility factors.

**Table A1.** Globally integrated NPP and EP values. Percentage changes from 70ka-control experiment are provided in brackets.

| Experiment | Global NPP (Pg Cyr$^{-1}$) | Global Export Production at 177.5m depth (Pg Cyr$^{-1}$) |
|---|---|---|
| PI-control (pife1%) | 47.45 | 7.19 |
| 70ka-control (pife1%) | 42.81 | 6.98 |
| lambfe1% | 41.40 (-3.2%) | 7.02 (+0.6%) |
| lambfe3% | 39.93 (-6.7%) | 7.07 (+1.3%) |
| lambfe5% | 39.44 (-7.8%) | 7.11 (+1.9%) |
| lambfe7% | 38.78 (-9.4%) | 7.09 (+1.6%) |
| lambfe10% | 38.49 (-10%) | 7.12 (+2%) |
| lambfe20% | 37.97 (-11%) | 7.16 (+2.6%) |
| glacfe1% | 41.06 (-4%) | 7.00 (+0.3%) |
| glacfe3% | 39.32 (-8.1%) | 7.03 (+0.7%) |
| glacfe10% | 38.09 (-11%) | 7.07 (+1.3%) |
| glacfe20% | 37.67 (-12%) | 7.06 (+1%) |

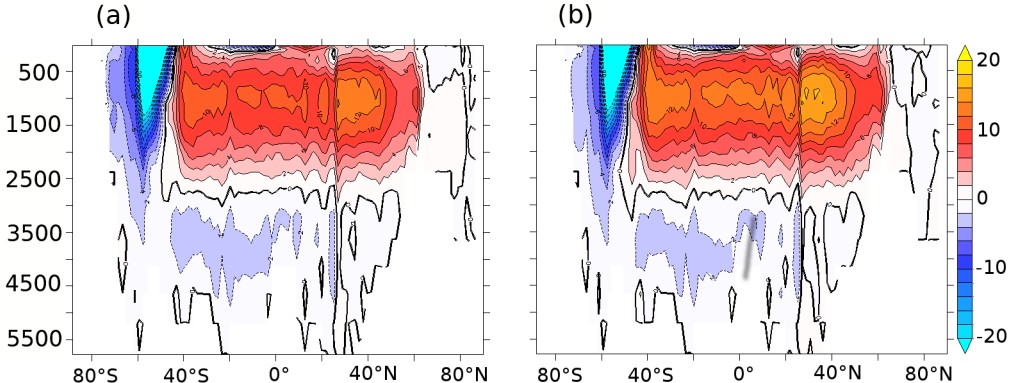

**Figure A3.** Atlantic meridional streamfunction (Sv) at (a) 70 ka BP and (b) PI

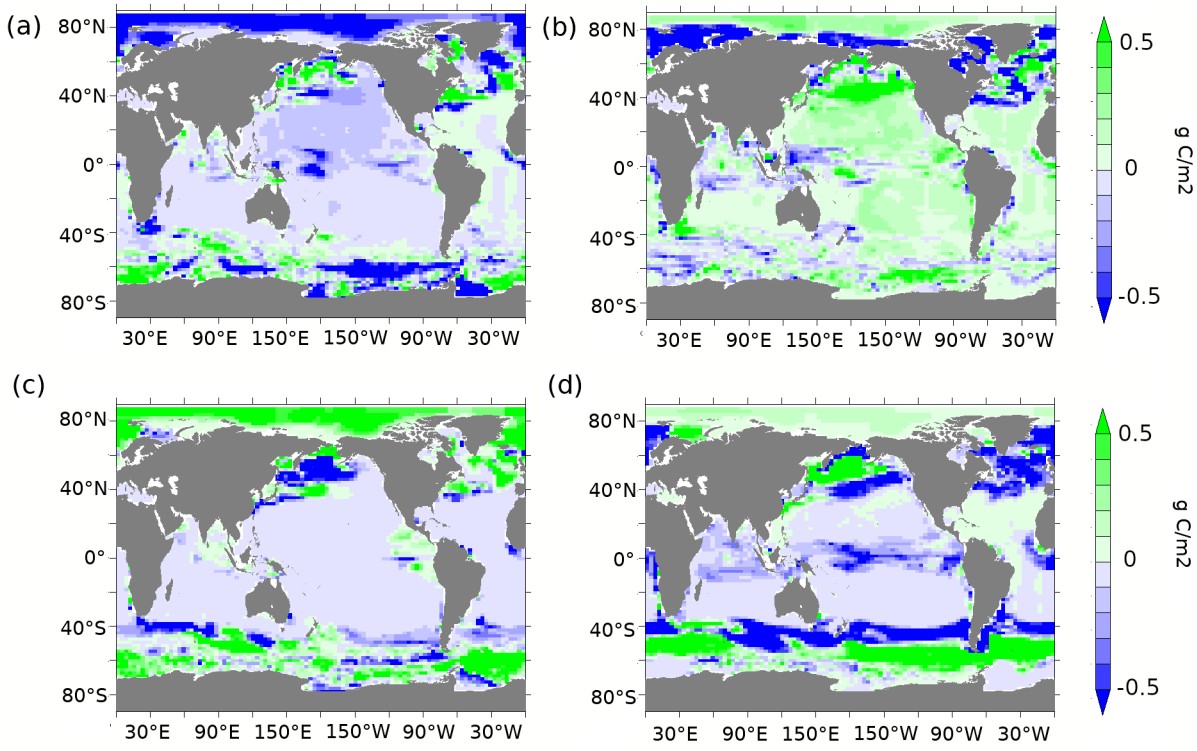

**Figure A4.** (a) Diatoms and (b) coccolithophores abundance anomalies (gC m$^{-2}$) at 70 ka compared to PI. (c) Diatoms and (d) coccolithophores abundance anomalies (gC m$^{-2}$) in lambfe3% compared to 70ka-control.

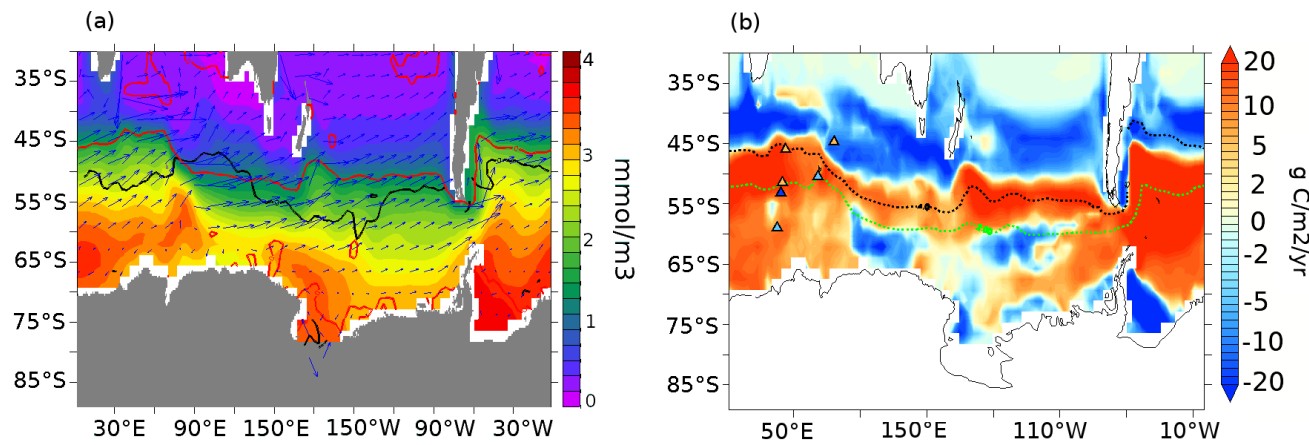

**Figure A5.** (a) Surface nitrate concentrations (mmol m$^{-3}$) for the 70ka-control experiment. The overlaid contours show the zero isoline of annual wind stress curl (black) and the zero contour of EP anomalies (red) between lambfe3% and 70ka-control (pife1%). Blue arrows represent surface currents; (b) Export production anomalies (gC m$^{-2}$ yr$^{-1}$) at 177.5m depth for lambfe3% compared to PI-control (pife1%) with opal flux (triangles) proxy records from (Amsler et al., 2022) (left of 50°E) and from (Thöle et al., 2019) (right of 50°E) to show qualitative comparison between model and data. Dark (light) orange represents significantly higher (slightly higher) and dark (light) blue represents significantly lower (slightly lower) values at 70ka compared to PI. Overlaid dashed contours represent modern SAF in black and APF in green based on the definition of Sokolov and Rintoul (2009).

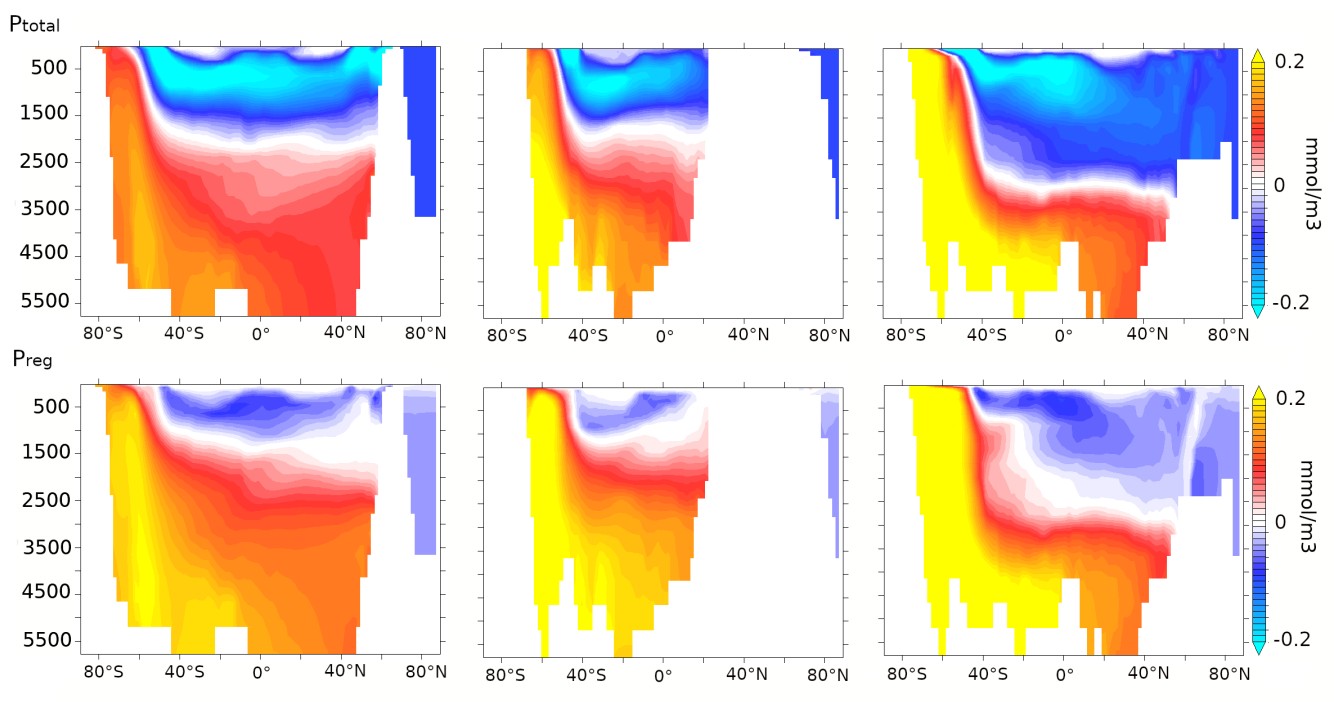

**Figure A6.** lambfe3% to 70ka-control (pife1%) anomalies of zonally averaged (top panels) total phosphate and (bottom panels) regenerated phosphate concentration anomalies (mmol m$^{-3}$) over (left) the Pacific, (center) Indian and (right) Atlantic basins.

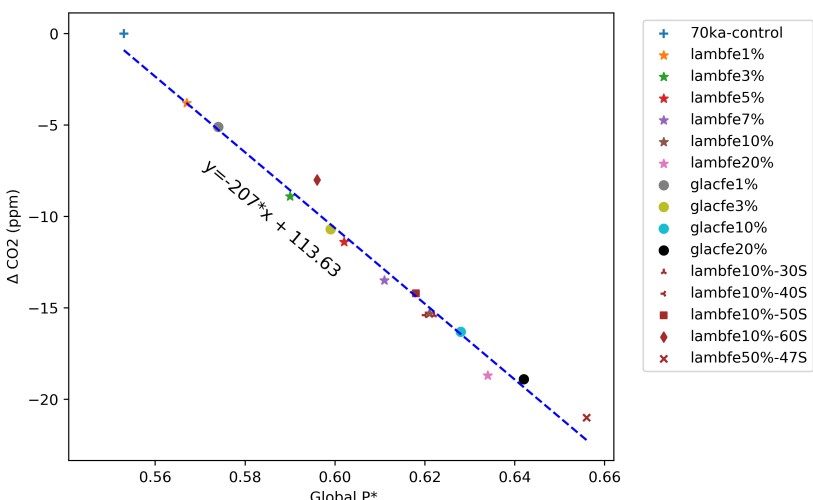

**Figure A7.** Atmospheric $CO_2$ (ppm) anomalies as a function of global P*. Slope= -207 ppm (compared to -312 ppm as per (Ito and Follows, 2005)).

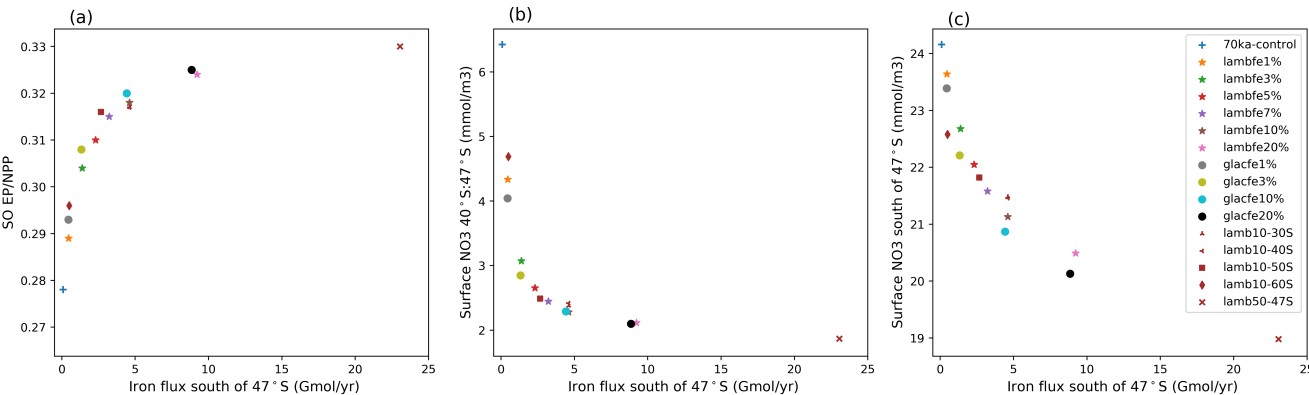

**Figure A8.** (a) Ratio of EP to NPP averaged over the Southern Ocean (south of $30°$S), (b) surface nitrate concentration averaged over $40°$S:$47°$S (mmol m$^{-3}$), and (c) surface nitrate concentration south of $47°$S (mmol m$^{-3}$) as a function of the aeolian iron flux (Gmol yr$^{-1}$) into the Southern Ocean south of $47°$S.

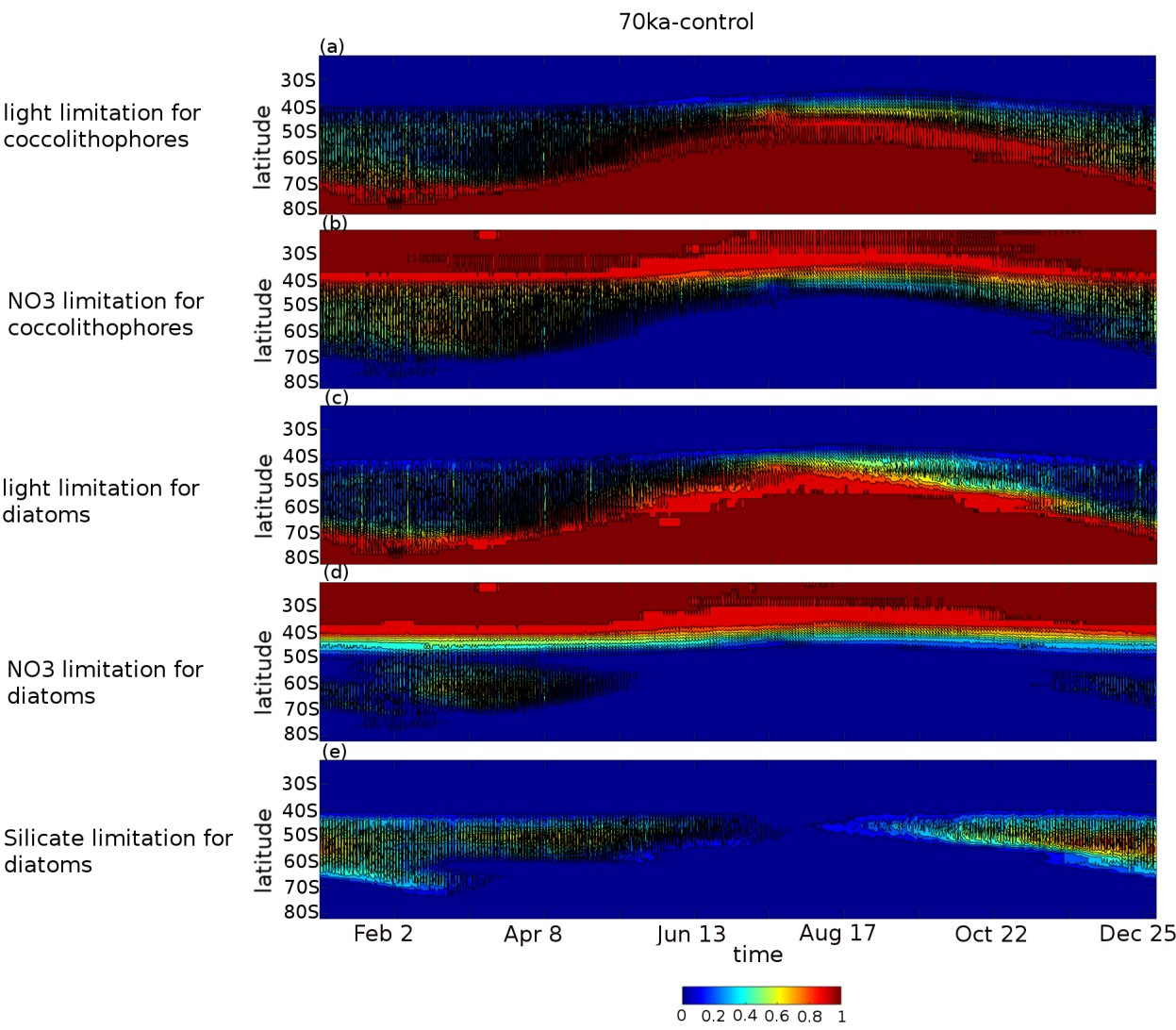

**Figure A9.** Hovmöller diagrams of the proportion of ocean grid cells per latitude band for the 70ka-control experiment for which (a) light is limiting coccolithophore growth (b) nitrate is limiting coccolithophore growth, (c) light is limiting diatom growth (d) nitrate is limiting diatom growth, (e) silicate is limiting diatom growth. In all the subpanels, if shade=1, then all ocean grid cells at that latitude band are limited by the respective limiting factor, if shade=0.5, then half of them are.

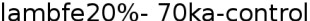

**Figure A10.** Hovmöller diagrams of the lambfe20% minus 70ka-control anomaly in the proportion of ocean grid cells per latitude band for which (a) nitrate is limiting coccolithophore growth, (b) nitrate is limiting diatom growth, (c) silicate is limiting diatom growth, (d) either of the macronutrients is limiting coccolithophore growth and (e) either of the macronutrients is limiting diatom growth.

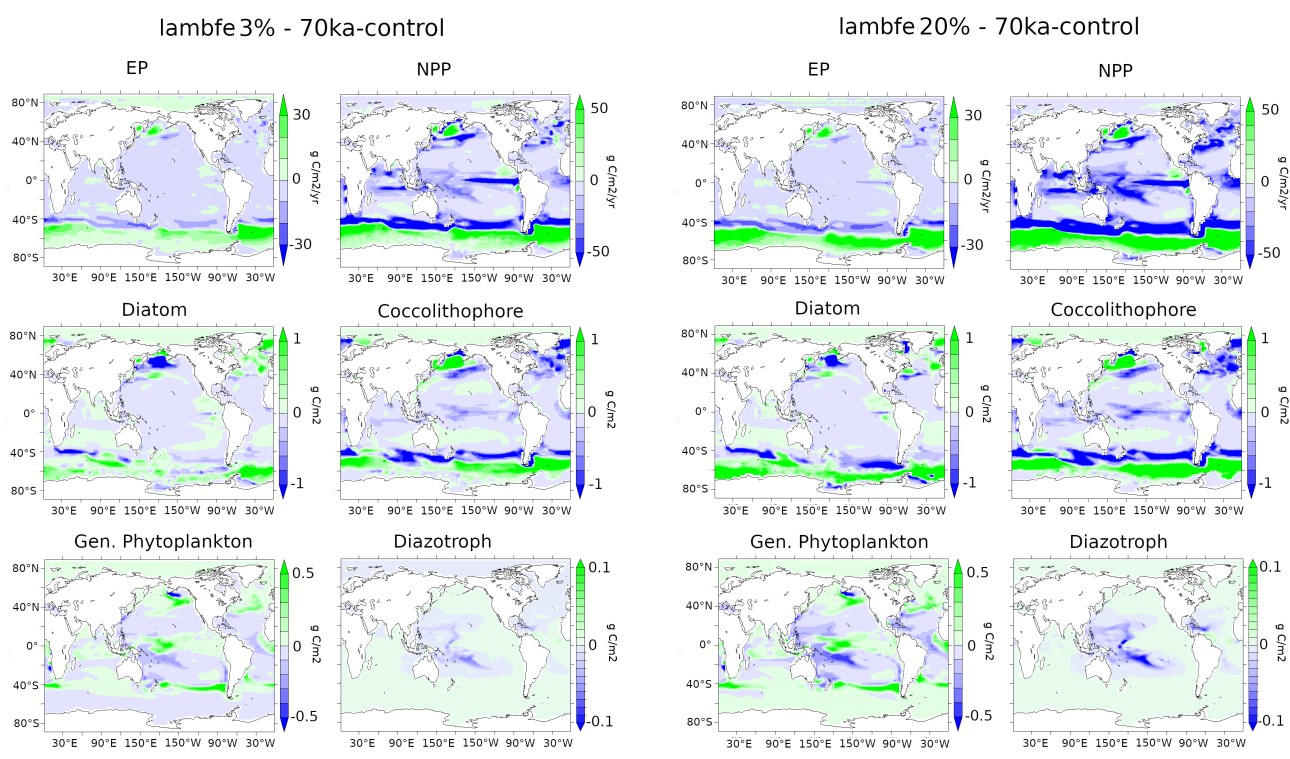

**Figure A11.** Ecosystem anomalies between lambfe3%-70ka-control and lambfe20%-70ka-control including EP (gC m$^{-2}$yr$^{-1}$) at 177.5m, NPP (gC m$^{-2}$yr$^{-1}$), diatoms (gC m$^{-2}$), coccolithophores (gC m$^{-2}$), general phytoplanktons (gC m$^{-2}$), and diazotrophs (gC m$^{-2}$).

*Data availability.* All the final data from modelling simulations is published on UNSW ResData repository (https://doi.org/10.26190/unsworks/24072)

*Author contributions.* HS performed all the simulations and analyses. KJM and LM supervised the research and assisted in the interpretation of the results. HS, LM and KJM drafted the manuscript. KK provided scientific support for the KMBM3 model and comments on an advanced 420 draft of the manuscript.

*Competing interests.* The authors declare that they have no conflict of interest.

*Acknowledgements.* Himadri Saini acknowledges funding from the University International Postgraduate Award scheme (UNSW). KJM and LM acknowledge support from the Australian Research Council (DP180100048, DP180102357, SR200100008 and FT180100606). KK is supported by the Global Change Through Time Programme administered by the New Zealand Ministry of Business, Innovation and 425 Employment. All experiments were performed on the computational facility of NCI owned by the Australian National University through awards under the Merit Allocation Scheme and the UNSW HPC at NCI Scheme.

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
