# Peer review of "Impact of iron fertilisation on atmospheric CO2 during the last glaciation"

_Climate of the Past, 2022_

## Author Comment (AC1)

The manuscript by Saini et al. is a continuation of a series of works where the effect of glacial-interglacial changes on planktic functional types and the carbon cycle is explored. Here they focus on differences between a portion of the onset of the last glaciacion (70 ky) and the Holocene. In particular the effect of different surface iron fluxes to the ocean on atmospheric CO2 and global ocean DIC is tested. The manuscript is clearly written. I have some comments regarding the methodology and presentation of results that should be addressed for the paper to be published by Climate of the Past.

We thank Dr Muglia for his helpful feedback on our manuscript. Please find below answers to the comments raised.

Black: Reviewer's comments.
Red: Author's responses
Green: Modified text in the manuscript.

In the methodology, the authors give an analysis of different estimates of iron from dust solubilities, and they define a "most likely range" between 3 and 5%. Is this estimate also valid for preindustrial dust? For your preindustrial simulation you pragmatically chose 1% solubility. The estimate of CO2 effect of Southern Ocean fertilization is based on the comparison of 70 ky simulations that use 3-5 % Fe solubility in the Southern Ocean vs a 70 ky simulation that uses preindustrial iron fluxes with 1% solubility. Due to the high variations in estimates of Fe solubility in the ocean, which the authors correctly point out, I think that the case of same solubility between glacial and interglacials should be included in the "most likely scenarios". This will affect the minimum of your range of CO2 change estimates due to Fe fertilization of the Southern Ocean.

We thank Dr Muglia for this very constructive comment and agree with the suggestion of including estimates of $CO_2$ drawdown based on the same solubility for glacial and interglacial dust. We have now included five new sensitivity experiments performed under 70 ka background conditions but using a pre-industrial (PI) iron mask with an iron solubility factor ranging between 3% and 20%. This approach allows us to estimate the minimum change in $CO_2$ drawdown due to glacial dust fluxes, assuming no change in solubility over time. We have included this information in the text as below:

Lines 135-143:

However, due to the uncertainties associated with present-day iron solubilities, we perform additional sensitivity experiments under 70 ka BP boundary conditions using the PI iron dust mask with iron solubility varying between 3% and 20%. This approach allows us to estimate the minimum change in $CO_2$ due to glacial dust fluxes, assuming no change in solubility over time. The corresponding $CO_2$ changes can be calculated by taking the difference between $CO_2$ changes achieved with the full experiments (i.e., changing masks and solubilities) and the $CO_2$ changes achieved by only changing solubility. This approach was validated by performing two additional 70ka equilibrium experiments with the pife mask and an iron solubility of 3% and 10% from which we branched off simulations with the lambfe mask and constant solubility (not shown). The resulting $CO_2$ drawdown in these experiments was the same than if calculated as the difference between the full experiments and solubility-only experiments.

The results from these new experiments are now included throughout the manuscript.

Lines 9-10:

If surface water iron solubility is considered constant through time, we find a $CO_2$ draw-down of ~4 to ~8 ppm.

Lines 198-200:

Changing the iron flux masks from PI to glacial at 70ka BP leads to a 3.8 to 8.3 ppm drop in atmospheric $CO_2$ concentration if we assume that the mean iron solubility remains unchanged (Table 2). Interestingly, for solubilities of 3% and higher, the drawdown is nearly constant, regardless of the glacial dust flux mask and regardless of the solubility ($7.3 \pm 1$ ppm).

Lines 363-369:

We find that the biological response to changes in iron fertilization not only depends on the iron solubility during glacial periods but also on the iron solubility during warm periods. Our results are based on the assumption that the global average iron solubility during warm periods equals ~1%. At higher initial values, the total potential draw-down of $CO_2$ would be smaller. For example, for an assumed solubility during warm periods closer to ~3%, we simulate a range of $CO_2$ changes between 6.4 ppm (no change in solubility and glacial fluxes based on Lambert et al., 2015) and 16.4 ppm (change to 20% solubility and glacial fluxes based on Ohgaito et al., (2018). This range reduces to 6.9-14.4 ppm if the initial solubility was 5% and to 6.7-6.9ppm if the initial solubility was 20%.

However, there is evidence suggesting that iron solubility could have been higher during glacial than interglacial periods (Lines 124-129). We now add the following text:

Lines 129-131:

[revised manuscript text omitted]

The difference in results between using either of the two glacial dust flux estimates that you apply in your experiments appears to give a weaker response than changing the solubility. This is an important result, which means that the global carbon cycle is more sensitive to factor changes in soluble iron flux that to differences in the horizontal flux pattern. It would be a good idea to discuss the result in the paper.

We thank the reviewer for this comment. Indeed, the changes in solubility factor have a larger impact on atmospheric $CO_2$ than the dust flux pattern in the Southern Ocean. We have indicated this result in L. 190-195. We also include this in the discussion as below:

Lines 345-348:

Despite these differences in spatial patterns, the two iron masks (glacfe and lambfe) with the same iron solubilities lead to similar decreases in atmospheric $CO_2$ in our simulations. This indicates that changes in atmospheric $CO_2$ are more dependent on changes in solubility, than on regional differences in aeolian iron fluxes in the Southern Ocean.

One of the novelties of the methodology is that it includes four functional types of phytoplankton: Normal phytoplankton, diazotrophs, coccolithofores, and diatoms. But how much does the inclusion of the new functional types affect the CO2 drop in the Southern Ocean fertilization experiments? In other words, how do the results of Saini et al. compare with a model with only regular phytoplankton and diazotrophs? It would be a benefit to the scientific community to have that comparison documented, since it would give us a hint on how necessary it is to include more planktic functional types in glacia-interglacial experiments.

We agree with the reviewer's comment that it is important to understand how the inclusion of different functional types of phytoplankton impacts the atmospheric $CO_2$ decrease. Lines 350-361 include a discussion of our results in comparison with previous modelling studies, which use different biogeochemical models. It should however be noted that all these previous modelling studies, except Menviel et al., 2012, were performed under different climate boundary conditions. Due to the different climate models and different boundary conditions, the climate changes differ among the simulations, thus making it challenging to isolate the impacts of the inclusion of different plankton classes. Nevertheless, we have now added the below paragraph:

Lines 381-387

None of the previous modelling studies simulate coccolithophores and diatoms' abundances prognostically. By including four distinct classes of plankton in our model we highlight the competitive dynamics between different major phytoplankton functional types for light and nutrient availability. Coccolithophores contribute to the total carbon export mainly in the polar frontal zone, while diatoms' contribution is in the Antarctic zone. As previously mentioned, carbon export close to convection sites in the Southern Ocean can be more efficient in reducing atmospheric $CO_2$. Furthermore, while both diatom and coccolithophores contribute to $CO_2$ uptake in the ocean through photosynthesis, coccolithophores produce $CaCO_3$ rich platelets, which reduce surface ocean alkalinity, thus reducing the $CO_2$ uptake efficiency.

What's the CO2 difference between your 70 ky and preindustrial experiments? I imagine it is higher in the 70 ky experiment due to lower global EP. Is that effect overcompensated by the Southern Ocean fertilization effect? It would be important to note this in the paper, since in the real climatr system the onset of the glaciation included physical changes as well as the assumed iron fertilization.

Both 70ka-control and PI runs were equilibrated with constant (forced) $CO_2$ values. The $CO_2$ is set at 289.5 ppm for PI and at 222.5 ppm for 70 ka based on the composite of Antarctic ice core estimates (Bereiter et al., 2015). Once the carbon cycle equilibrated to these imposed $CO_2$ concentrations, we calculated $CO_2$ prognostically for a few hundred years, to ensure that there is no remaining drift. While it would be interesting to study the impact of climate change on the 70ka $CO_2$ level, this is out of the scope of the present study.

Another novelty of the model is the inclusion of a sediment model, which allows changes in global alkalinity. Does global alkalinity vary significantly among your experiments? Does the variation in CO2 that you find depend on global alkalinity changes at all? This is a topic that could be potentially addressed in this paper.

We thank the reviewer for this comment and agree that changes in oceanic alkalinity can significantly impact atmospheric $CO_2$. However, in our experimental setup, the river inflow compensates for the $CaCO_3$ burial to keep global alkalinity constant. This was a choice we made at the beginning of this project, and although we agree that this would be an interesting question to explore, we cannot do so with our simulations.

Minor comments:

Line 19: No need for the ~ symbol if you include uncertainty.

Incorporated throughout the text.

Line 121: What leads? The sentence is not clear.

The lambfe aeolian iron input south of 47S with a 50% solubility is equivalent to 23.05 Gmol/yr iron input. The sentence is now modified as:

Lines 148-150:

In the fifth sensitivity experiment (lambfe50%-47S), the aeolian iron input south of 47°S follows the lambfe mask with a solubility factor of 50% (Figure A1f), which is equivalent to 23.05 Gmolyr$^{-1}$ iron input in the Southern Ocean south of 47°S and provides an upper limit on the potential $CO_2$ draw-down.

Fig. 2: Color scale poorly chosen, it saturates in most places.

We apologise for the poor choice of colour scale. Figure 2 in the manuscript is now modified and shown as Fig. R2 below:

[Figure]

Fig.R2: Aeolian iron dust flux (μmol m$^{-2}$ yr$^{-1}$) anomalies for (a) lambfe1% minus pife1% and (b) glacfe1% minus pife1%.

Fig. 3: Why does the North Altantic show such a complex pattern of EP anomalies? Is it changes in convection, ventilation, the AMOC?

Yes, the complex changes are due to shifts in convection sites in the North Atlantic. A main convection site south of Iceland in our PI simulation shifted to the east (please refer to fig. below), leading to lower diatom and higher coccolithophore populations at this location and an increase in diatoms in the eastern North Atlantic. In the west, near the coast of Nova Scotia, both diatoms and coccolithophores increase.

[Figure]

Fig.R3: Ventilation depth (m) anomalies for 70ka-control compared to PI.

The following text is added as below:

Lines 187-188

The North Atlantic shows a complex pattern of anomalies due to changes in the strength and location of deep ocean convection, which result in an overall decrease in NPP and EP by 16% and 9%, respectively.

Fig. 4d shading colors not easily distinguishable. Why does terrestrial carbon go down in your simulations?

We have modified the colour scale of Figure 4d (as also shown in Fig.R1 above).

As the iron input to the ocean increases, so does the efficiency of the biological pump, which leads to an increase in the oceanic carbon reservoir and thus an atmospheric $CO_2$ decrease. The atmospheric $CO_2$ decrease in turn leads to a cooling, and overall drier conditions over land. This climate change reduces the terrestrial carbon reservoir, with contributions from both soil and vegetation carbon.

This explanation is given in lines 255-259:

The simulated decrease in atmospheric $CO_2$, equivalent to 8-45 GtC (Figure 4d, red shade), leads to a decrease in surface air temperatures, as well as regional changes in precipitation and soil moisture. In addition, the lower atmospheric $CO_2$ concentration also reduces photosynthesis and consequently litter fall. The direct and indirect effects of a lower atmospheric $CO_2$ concentration result in a terrestrial carbon decrease of 16 to 88 GtC (Figure 4d, green shade), out of which 8 to 45 GtC decrease is from terrestrial vegetation while 8 to 43 GtC reduction is from soil carbon.

[Figure]

Fig R4: Soil (left) and vegetation (right) carbon (g/m2) anomalies for lambfe3% compared to 70ka-control.

The global iron flux in experiments lambfe10% and lambfe3% is not shown anywhere. The fluxes should be plotted somewhere in the paper, since the results of these simulations are discussed in detail.

We apologise for this missing information. In Fig.2 in the manuscript, we show the lambfe and glacfe 1% dust fluxes compared to pife 1% flux. For completeness we now also show pife, lambfe, glacfe iron fluxes with 3% and 10% solubility. The fluxes are now added in the supplementary as Figure A2.

[Figure]

Fig. R5: Aeolian iron dust fluxes (µmol m$^{-2}$ yr$^{-1}$) for pife (a,d), lambfe (b,e) and glacfe (c,f) masks with 3% (top row) and 10% (bottom row) solubility factors.

---

## Author Comment (AC2)

Comment for "Impact of iron fertilisation on atmospheric $CO_2$ during the last glaciation"

Saini and co-author present nice modelling study of glacial carbon changes at 70 ka BP. The authors investigate, using a range of factorial analyses, the impacts of iron input and increased efficiency of biological pump in the glacial ocean. They simulate an upper limit for the CO2 decrease due to iron fertilization of 21 ppm. The manuscript is concise and well written. Figures and tables are illustrative and support the conclusions. I recommend publication of the manuscript after minor revision.

We thank the reviewer for their positive appraisal of the manuscript. The following comments have now been addressed.

I have one concern that require revision. The author showed that an exponential decay response of $CO_2$ decrease to iron input to the Southern Ocean (SO) is due to the saturation of the biological pump. I agree with this conclusion, but some descriptions of the reason why the efficiency of the biological pump saturates in response to iron inputs are missing. As a result of the saturation of the increase in EP/NPP in the SO in response to the iron input, has the efficiency of the biological pump also saturated? For example, please add figures for changes in EP/NPP in SO in response to iron inputs in SO to Figure 4 and describe the nonlinear response of the ecosystem to changes in iron inputs.

We thank the reviewer for this comment. We have taken the helpful suggestion of including a scatter plot of EP/NPP to show the non-linear response of Southern Ocean ecosystems to iron fertilisation. For low iron levels, EP/NPP increases sharply with increasing iron. However, at high iron levels, further iron input only increases EP/NPP marginally (Fig.R1a, now included in the Appendix as Figure A8a).

The saturation of the efficiency of the biological pump is also indicated by changes in P* as a function of iron input (Figure 4c). P* represents the ratio of regenerated phosphate over total phosphate. Fig. 4c shows that at high iron levels, further iron input only marginally impacts P*. The explanation behind the saturation in P* is included in the modified text at the end of the response to this comment.

[Figure]

Fig.R1 (a) Ratio of EP to NPP averaged over the Southern Ocean (south of 30°S), (b) surface nitrate concentration averaged over 40°S:47°S (mmol m$^{-3}$), and (c) surface nitrate concentration south of 47°S (mmol m$^{-3}$) as a function of the aeolian iron flux (Gmol yr$^{-1}$) into the Southern Ocean south of 47°S.

It would also be important to add a similar figure for changes in nutrient concentrations to show changes in nutrient cycling with iron fertilization. In the case of large iron flux (e.g. lambre50%-47S), does the ecosystem response saturate due to nitrogen depletion instead of iron? I think these descriptions would be useful to the reader.

We are now including an analysis of the factors limiting the growth rate of diatoms and coccolithophores in the Southern Ocean.

Photosynthesis in this model is calculated by multiplying the concentration of the phytoplankton functional type in question with the minimum of three/four terms which are light, nitrate and phosphate for coccolithophores, and additionally silicate for diatoms (Equations 13-16 in Kvale et al., 2021). Please note that nitrate limitation depends on nitrate concentration scaled by iron availability and ocean temperature (Equations 3 and 5 in Kvale et al., 2021). The same is true for phosphate and silicate limitation. The light limitation includes light attenuation by plankton biomass and sea ice cover (Equation 12 in Kvale et al., 2021) and is also scaled by temperature and iron availability (Equation 9 in Kvale et al., 2021). In Fig. R2 and R3 we show Hovmöller diagrams showing the seasonal evolution of the proportion of ocean grid cells at each latitude band that are limited by each of these factors for diatoms and coccolithophores in our simulations.

At low iron levels (our 70ka-control run, Fig. R2, added as Figure A9 in the manuscript), light limits the growth of coccolithophores and diatoms south of ~75°S throughout the year (Fig. R2a, c), while nitrate limits the growth of these plankton functional types in more than 80% of the cells north of ~40°S (Fig. R2b, d). In the band between ~40°S and ~75°S, the limitation moves back and forth between light and nitrate for coccolithophore growth (Fig. R2a, b) and between light, nitrate, and silicate for diatom growth (Fig. R2c-e). Phosphate only limits the plankton growth in the 70ka-control run for a very short time at the northern boundary of our region of interest (~20°S). We therefore concentrate on nitrate and silicate in our analysis.

[Figure]

Fig.R2: Hovmöller diagrams of the proportion of ocean grid cells per latitude band for the 70ka-control experiment for which (a) light is limiting coccolithophore growth (b) nitrate is limiting coccolithophore growth, (c) light is limiting diatom growth (d) nitrate is limiting diatom growth, (e) silicate is limiting diatom growth. In all the subpanels, if shade=1, then all ocean grid cells at that latitude band are limited by the respective limiting factor, if shade=0.5, then half of them are.

To illustrate the changes occurring as more iron is added into the Southern Ocean, we show the changes in the proportion of ocean cells limited by each factor in experiment lambfe20% compared to the 70ka-control (Fig. R3, added as Figure A10 in the manuscript). The region between 40°S and 50°S becomes significantly more nitrate limited for both coccolithophores and diatoms in the lambfe20% experiment (Fig. R3a, b). With higher iron availability, NPP and export production south of 47°S increases, using nutrients more efficiently (Fig. R1c, now added as Figure A9c in the appendix) and reducing the northward nutrient advection (Fig. R1b, now added as Figure A9b in the appendix). At the same time silicate limitation for diatoms decreases in this region (Fig. R3c). This is because, as the iron flux increases, both coccolithophores and diatoms shift southward leading to a decrease in diatoms between 40°S and 55°S. This further enhances silica availability in this region, consequently leading to a decrease in silicate limitation.

We also show the proportion of ocean cells that are limited by either of the macronutrients (nitrate or phosphate for coccolithophores; and nitrate, phosphate, or silicate for diatoms) in Fig. R3d and e.

In this figure we see that the overall nutrient limitation between ~40°S and 50°S increases in the high iron flux experiment compared to 70ka-control.

Therefore, in agreement with the reviewer's comment, we now state that the saturation in the biological pump is indeed due to nitrate limitation in the northern band of the Southern Ocean as the iron fluxes increase.

[Figure]

Fig.R3: Hovmöller diagrams of the lambfe20% minus 70ka-control anomaly in the proportion of ocean grid cells per latitude band for which (a) nitrate is limiting coccolithophore growth, (b) nitrate is limiting diatom growth, (c) silicate is limiting diatom growth, (d) either of the macronutrients is limiting coccolithophore growth and (e) either of the macronutrients is limiting diatom growth.

We now include this information as the following text and as Figures A9 and A10 respectively.

Lines 244-253

However, as the Southern Ocean iron flux increases, the overall iron fertilisation efficiency reduces. While export production increases south of 47°S, thus using nutrients more efficiently (Figure A8c) and reducing the nutrient advection north of the SAF (Figure A8b), nitrate limitation increases in the sub-Antarctic zone (Figure A9b, d and Figure A10a, b), leading to a decrease in export production. For example, in experiment lambfe20%, a 67% decrease in nitrate concentration is simulated between 40°S and 47°S (Figure A8b). At the same time, silicate limitation decreases in the sub-Antarctic zone (Figure 9e and Figure A10c) due to a southward shift of both coccolithophores and diatoms and a decrease in diatoms between 40°S and 55°S (Figure A11). This further enhances silica availability in this region, consequently leading to a decrease in silicate limitation. Therefore, the total biological pump efficiency, represented here by changes in P* (Figure 4c) and Southern Ocean EP to NPP ratio (Figure A8a), saturate at high iron values due to nitrate limitation north of 50°S in the Southern Ocean. The global P* in lambfe20% and in lambfe50%-47S (Figure 4c) equal 0.63 and 0.65 respectively, suggesting a maximum efficiency of 65% in our experimental set-up.

Minor comments

L174-175: It is interesting result that even though the two iron deposition patterns are very different, the effect on $CO_2$ lowering is almost the same for the two patterns. Please add an explanation as to why the Weddell Sea convection would result in smaller changes in $CO_2$ for the different iron deposition patterns.

We thank the reviewer for this suggestion. Deep convection in the Weddell Sea makes this region slightly more efficient at drawing down atmospheric $CO_2$. We have added the following sentences:

 Lines 214-219:

The glacfe mask is however slightly more efficient in drawing down atmospheric $CO_2$. The iron flux into the South Atlantic sector is higher in the glacfe mask compared to the lambfe mask, increasing iron concentrations near one of our major convection sites in the Weddell Sea. Bottom water formation in this region leads to greater mixing with deeper layers, replenishing surface nutrient concentrations and resulting in higher export production (Marinov et al., 2006). Global P* is ~1.5% higher in our glacfe experiments than in our lambfe experiments resulting in a slightly larger $CO_2$ drawdown.

We are also making this clearer in the discussion.

Lines 348-351:

The experiments using the glacfe mask are slightly more efficient in drawing down $CO_2$. One of the major Antarctic Bottom Water formation sites is located in the Weddell Sea in our simulations, making the South Atlantic sector a more efficient region for carbon sequestration. Higher iron dust input in the Weddell Sea in the glacfe experiments thus leads to greater $CO_2$ drawdown compared to the lambfe experiments.

L286-287 Where is this information described in the method? I couldn't figure it out, so please let me know.

This information is described in Section 2.2, 1st paragraph.

---

## Author Comment (AC3)

Saini et al. present an interesting modeling study in which the authors investigate the response of the global marine ecosystem to an increase in the supply of dust - and by inference Fe – to the ocean surface at the last glacial inception, 70 ka. The scarcity of Fe is indeed considered to limit biological/export production in large swaths of the world ocean, in particular in polar regions today and in the past.

Their sensitivity experiments indicate that increased availability of Fe conduces to a strengthening of the marine biological carbon pump, which contributes to sequester C in the ocean interior, decreasing atmospheric CO2 concentrations (4-16 ppm). What's more, their model outputs suggest that most of the action is centered in the Southern Ocean (SO). Furthermore, the experiments suggest that the strengthening of the biological carbon pump consequent to Fe fertilization rapidly saturates as a result of complex interactions involving downstream effects, to me, the most interesting conclusion of the work.

The study is certainly interesting and worth publishing, yet I feel is it not sufficiently anchored in available palaeoceanographic records. High-resolution EP reconstructions across the MIS4/MIS5 interval are arguably limited, yet available records are often at odds with the model outputs - notably in the subsarctic North Pacific and the SO - which warrants a much more detailed/nuanced discussion. What's more, given the sensitivity of the SO to increased supply of dust/Fe, I would urge the authors to distinguish the SAZ and AZ, as these regions have responded very differently in terms of EP.

We thank the reviewer for their very interesting comments and valuable sources of information for proxy records. We agree that our iron fertilisation discussion should distinguish the responses of the sub-Antarctic zone (SAZ) and Antarctic zone (AZ) in the Southern Ocean. We have now incorporated this suggestion throughout the manuscript and have also provided detailed comparison with the suggested proxy records. As the focus of this paper is on Southern Ocean iron fertilisation and its impact on global $CO_2$ changes, we do not present a detailed discussion of the changes occurring in the North Pacific. Nevertheless, we have mentioned the differences between our results and the proxy data available in this region.

Please find my comments below, which I hope the authors will find constructive.

l. 39 - Interestingly, d13CO2 data suggest that the drop in atmospheric CO2 centered around 70 kyrs was likely related to a combination of factors including enhanced C storage in the ocean subsurface and decreased air-sea gas exchange in the SO (Eggleston et al., 2016; Menking et al., 2022).

We thank the reviewer for referring these studies. We are now including $\delta^{13}CO_2$ records from Eggleston et al., (2016) and Menking et al. (2022) in Figure 1 (Fig. R2 below) and are discussing their implication in the introduction:

Lines 42-47:

Interestingly, high resolution $\delta^{13}CO_2$ records from Antarctic ice cores (Menking et al., 2022) display a 0.5 permil decrease centred at 70.5 ka, followed by a 0.7 permil increase (Figure R2g), indicating a complex set of processes impacting atmospheric $CO_2$ at the MIS5-4 transition. While surface ocean cooling could explain the concurrent $CO_2$ and $\delta^{13}CO_2$ decrease, the $\delta^{13}CO_2$ increase in the second part of the transition would be consistent with a greater efficiency of the biological pump and increased storage of respired carbon in the deep ocean (Menviel et al., 2015, Eggleston et al., 2016, Menking et al., 2022).

l. 49-51 & 227-233 - I find this argument somewhat misleading. I would encourage the authors to discuss the SAZ and AZ separately, as mentioned above. Palaeoceanographic data indeed suggest an increase in EP in the SAZ at the MIS4/5 transition (e.g. Martinez-Garcia et al., 2011/2014; Lamy et al., 2014; Thöle et al., 2019; Amsler et al., 2022). However, palaeoceanographic records from the AZ suggest a simultaneous decrease in EP (e.g. Anderson et al., 2009; Jaccard et al., 2013; Studer et al., 2015; Thöle et al., 2019; Amsler et al., 2022).

We thank the reviewer for this comment. In our model, the 47°S boundary roughly matches with the subantarctic front (SAF) which delineates the subantarctic zone (SAZ) and the Antarctic zone (AZ). Our current description of the results is based on this boundary, north and south of which we see changes of opposite sign. This has been made clearer now using the terms SAF, SAZ and AZ throughout the manuscript.

Lines 55-59:

These peaks in dust flux are concurrent with increased export production (EP) in the subantarctic zone (SAZ) of the Southern Ocean during the LGM (Kohfeld et al., 2005, 2013; Martinez-Garcia et al., 2014), as well as during MIS4 (Lamy et al., 2014; Martinez-Garcia et al., 2014; Thöle et al., 2019; Amsler et al., 2022). On the other hand, palaeoceanographic data from the Antarctic zone (AZ) suggest a decrease in EP at the LGM and MIS4 (Kohfeld et al., 2005,2013; Anderson et al., 2009; Jaccard et al., 2013; Studer et al., 2015; Thöle et al., 2019; Amsler et al., 2022).

Lines 189-196:

Within the AZ (south of the APF), diatoms decrease by 14% in the Pacific, while they increase by 3% in the Atlantic and Indian sectors (Figure A4a). The decrease in diatoms in the Pacific sector of the AZ leads to a competitive growth advantage for coccolithophores, which increase by 6.5% south of the polar front, thus leading to a poleward shift of coccolithophores in the Pacific sector (Figure A4b). On the other hand, coccolithophores decrease in the Atlantic and Indian sectors of the AZ by 15% and 8%, respectively. Diatoms increase by 22% in the SAZ (north of the SAF) while there are no significant changes in coccolithophores abundance. As a result of the changes in these two plankton species, EP increases by 1.3% in the SAZ and decreases by 14% in the AZ (Figure 3b). The total Southern Ocean (south of 30°S) EP and NPP decrease by 2% and 7.5%, respectively at 70 ka.

Lines 225-227:

Our experiments show that the ecosystem response north and south of ~47°S (which is roughly the modern SAF) are of opposite sign (Table 3, Figure 6).

Lines 271-276:

Iron increase in the lambfe3% experiment leads to a 37% increase in diatoms and an 88% increase in coccolithophores in the AZ (Figure A4c, d). The simulated increased EP in the AZ (Figure 5a) leads to greater nutrient utilisation south of the APF (Figure A9c). On the contrary, because of lower nutrient availability, both diatoms and coccolithophores abundances decrease in the SAZ by 46% and 31% respectively (Figure A4c, d). Consequently, in the lambfe3% experiment, the EP increases by 98% in the AZ while it decreases by 17% in the SAZ compared to the 70ka-control simulation (Figure 5a, 6a).

We have also added the following detailed discussion on the model data discrepancy and Fig. R1 as Figure A5b in the appendix:

[Figure]

Fig.R1: (a) Surface nitrate concentrations (mmol m⁻³) for the 70ka-control experiment. The overlaid contours show the zero isoline of annual wind stress curl (black) and the zero contour of EP anomalies (red) between lambfe3% and 70ka-control (pife1%). Blue arrows represent surface currents; (b) Export production anomalies (gC m⁻² yr⁻¹) at 177.5m depth for lambfe3% compared to PI-control (pife1%) with opal flux (triangles) proxy records from Amsler et al., 2022 (left of 50°E) and from Thöle et al., 2019 (right of 50°E) to show qualitative comparison between model and data. Dark (light) orange represents significantly higher (slightly higher), and dark (light) blue represents significantly lower (slightly lower) values at 70ka compared to PI. Overlaid dashed contours represent modern SAF in black and APF in green based on the definition of Sokolov et al., (2009).

Lines 313-331:

Alkenone flux (Lamy et al., 2014; Martinez et al., 2014), as well as opal and organic carbon fluxes (Thöle et al., 2019) reconstructed from subantarctic sediment cores suggest that EP was higher in the SAZ during MIS4 than present day. In addition, bottom water oxygenation records indicate a deep ocean oxygenation decrease during MIS4, which might suggest increased respired carbon storage in the deep ocean (Amsler et al., 2022) but could also reflect a change in ocean dynamics and water residence time. It has been suggested that the subantarctic EP increase during MIS4 was due to higher iron fluxes into the Southern Ocean. Higher dust deposition in the South Pacific and the South Atlantic at 70 ka is consistent with available paleo-proxy records (Martinez et al., 2011, Lambert et al., 2012, Lamy et al., 2014, Thöle et al., 2019). At the same time, palaeoceanographic records from the AZ suggest a decrease in EP during MIS4 (Anderson et al., 2009; Jaccard et al., 2013; Studer et al., 2015; Thöle et al., 2019; Amsler et al., 2022).

In our 70 ka simulations with enhanced iron input, EP increases in the AZ and polar frontal zone (between SAF and APF) and decreases in the SAZ, in contrast with most paleo-proxy records (Figure A5b). However, there is evidence that greater iron flux within the seasonal sea ice zone has led to higher diatom concentrations (Abelmann et al. [2006, 2015]) and likely also EP in this region during glacial periods. In our study, as the iron input into the ocean increases, the nutrient utilisation increases in the AZ, consistent with the existing d¹⁵N records in the AZ at the MIS4/MIS5 transition (Studer et al., 2015; Ai et al., 2020). As a result, the nutrient advection into the SAZ is reduced, thus leading to an EP decrease in the SAZ. In addition, our simulated EP increase in the AZ during MIS4 compared to PI, and thus increase in regenerated carbon storage in the deep ocean, is consistent with some of the proxy records suggesting a decrease in deep ocean oxygenation in the AZ (Jaccard et al., 2016, Amsler et al., 2022). Our EP increase is also consistent with the increase in opal flux north of the APF (Amsler et al., 2022) at the MIS4 onset (Figure A5b).

Fig. 1 – maybe the authors could include ice core d13CO2 data (Eggleston et al., 2016; Menking et al., 2022) in the figure?

We thank the reviewer for this suggestion. Ice core $\delta^{13}CO_2$ data from Eggleston et al., (2016) and Menking et al., (2022) is now added in the Figure 1 in the manuscript as fig. R2. below

[Figure]

Fig.R2: Time series of (a) Antarctic temperature anomalies from present day (°C) (Jouzel et al., 2007), (b) atmospheric $CO_2$ concentration (ppm) (Bereiter et al., 2015), and (c) dust flux (mg/m$^2$/yr) (Lambert et al., 2012) as recorded in EPICA DOME C ice core; (d) iron dust accumulation rates (mg/m$^2$/yr) from ODP Site 177-1090 (Atlantic) Martinez et al., 2011), iron (%) records in the South Pacific from Lamy et al., (2014) at sites (e) PS75-076 and (f) PS75-059; (g) atmospheric $\delta^{13}CO_2$ (‰) as recorded in a composite of Antarctic ice cores (purple line, Eggleston et al., 2016) and high resolution records from Taylor Glacier ice cores (dashed cyan line, Menking et al., 2022). Shaded areas represent marine isotope stages 2 (MIS2: 27-19 ka BP) and 4 (MIS4: 71-59 ka BP).

l. 92/115 & Table 1 – could the authors provide some more context related to the dust flux estimates used in the simulations? The reason I'm asking is that the reconstructed dust/Fe fluxes to Antarctica based on ice cores measurements are much larger when compared to Fe fluxes quantified based on marine sediments. These differences may relate to different transport pathways and/or atmospheric dynamics over the SO and Antarctica during the last ice age.

The dust flux estimates from Lambert et al. (2015) are globally interpolated 2-d dust flux patterns using the unevenly distributed dust flux data points (including data from both, ice cores and marine sediments), most of them collated in the DIRTMAP (Dust Indicators and Records of Terrestrial and Marine Paleoenvironments) database. Their interpolation method assumes that the aerosol concentration in the air decreases exponentially from the source. On the other hand, dust flux estimates from Ohgaito et al., (2018) were generated by a model, assuming dry and unvegetated regions as dust sources and simulating global dust transport and deposition. Here, both glacfe and lambfe iron fluxes were obtained by mapping the iron content of dust using the estimates of Zhang et al., (2015). We have included this information in the method section.

Lines 103-112:

From this control simulation, we branch off a suite of sensitivity experiments with prognostic atmospheric $CO_2$, using two different glacial iron dust flux estimates. The first iron flux, lambfe, is obtained from the LGM dust estimates of Lambert et al., (2015). This dust flux estimate is calculated by performing a global 2-d interpolation on unevenly distributed LGM dust flux records, most of which are collated in the DIRTMAP (Dust Indicators and Records of Terrestrial and Marine Paleoenvironments) database (Kohfeld and Harrison, 2001; Maher et al., 2010). Their interpolation method assumes that the aerosol concentration in the air decreases exponentially from the source distance. The second iron flux, glacfe, is derived from a dust flux obtained with a model (LGMglac.a, Ohgaito et al., (2018)). This model includes glaciogenic dust sources in addition to the usual desert dust sources and assumes dry and unvegetated regions as dust sources. It then calculates global dust transport and deposition. Both glacfe and lambfe are then obtained in this study by mapping the iron percentage on dust (Zhang et al., 2015) as mentioned above.

l. 150 – is the AMOC both weaker and significantly shallower in the 70 kyr simulation?

As seen in Fig. R3 (now added as Fig. A3 in the appendix), the AMOC is not significantly shallower in the 70ka simulation compared to PI.

[Figure]

Fig. R2: Atlantic meridional overturning streamfunction (Sv) at (a) 70 ka BP and (b) PI.

We also add this information as below:

Line 178:

The simulated AMOC strength in the 70ka-control experiment is ~14Sv, compared to ~17Sv in the PI-control, without significant changes in its depth.

l. 154-156 & 234-236 - Interesting… but these observations seem at odds with paleoceanographic reconstructions, which generally show a decrease in EP at the MIS4/MIS5 transition (e.g. Jaccard et al., 2005; Gebhardt, et al., 2008). These observations are consistent with paleoceanographic reconstructions spanning the last deglaciation, which consistently indicate lower EP during the LGM, even though Fe supply might have been higher then (e.g. Kohfeld & Chase, 2011 for a review)

We thank the reviewer for this comment and providing us with these references. The simulated increase in EP in the North Pacific is due to greater $NO_3$ availability in our 70ka-control simulation; caused by higher denitrification in the North Pacific in our PI-control simulation. This leads to an increase in diatoms and coccolithophores in this region and thus EP.

However, as this study focuses on iron fertilisation in the Southern Ocean, we did not discuss results from the North Pacific in detail especially for differences between 70ka and PI climate which use the same iron masks and iron solubility factor. We now mention this model data discrepancy in the paragraph as below:

Lines 183-187:

The simulated 18% EP increase at 70 ka in the North Pacific (Figure 3b) can be attributed to greater diatom and coccolithophore abundance in that region (Figure A4a, b), resulting from higher nutrient availability. This EP increase in the North Pacific is however inconsistent with the biogenic Ba (Jaccard et al., 2005) and $d^{15}N$ (Gebhardt et al., 2008) records from the sub-arctic North Pacific which suggest a decrease in EP at MIS4 onset.

l. 153 – I'm not sure to understand what is meant by "strength of the deep water masses"?

We have modified this sentence as below:

The colder conditions, more extensive sea-ice cover, and changes in the deep ocean convection impact marine productivity (Saini et al., 2021).

l. 156 – what do you mean by "general phytoplankton"?

There are four different classes for phytoplankton in the ecosystem model used in this study. Three of which include specifically characterized plankton such as diazotrophs, that can fix nitrogen, coccolithophores, that produce $CaCO_3$ shells, and diatoms that produce opal. The fourth class is for the rest of the types of plankton, that are mostly located in the low latitude regions. This is now explicitly described in the manuscript:

Lines 84-87:

There are four different classes for phytoplankton in this ecosystem model. Three of which include specifically characterized plankton such as diazotrophs that can fix nitrogen, coccolithophores that produce $CaCO_3$ shells, and diatoms that produce opal. The fourth class is for the rest of the types of plankton, that are mostly located in the low latitude regions.

l. 157-163 – again, I would recommend discussing the SAZ and AZ separately.

We have now included this information as below:

Lines 173-176:

The strongest ocean surface cooling (-1.45°C) at 70 ka BP with respect to our PI-control is simulated north of 40°N in the Atlantic and Pacific oceans, while the annual mean SSTs in the SAZ and AZ are ~0.8°C and 0.4°C lower than in the PI-control, respectively (Figure 3a).

l. 168-169 – why is that? Are ecosystems rapidly becoming N-limited as Fe availability increases?

Yes, the ecosystem becomes nitrate limited in the northern band of the Southern Ocean as iron flux increases in our simulations.

We are now including an analysis of the factors limiting the growth rate of diatoms and coccolithophores in the Southern Ocean.

Photosynthesis in this model is calculated by multiplying the concentration of the phytoplankton functional type in question with the minimum of three/four terms which are light, nitrate and phosphate for coccolithophores, and additionally silicate for diatoms (Equations 13-16 in Kvale et al., 2021). Please note that nitrate limitation depends on nitrate concentration scaled by iron availability and ocean temperature (Equations 3 and 5 in Kvale et al., 2021). The same is true for phosphate and silicate limitation. The light limitation includes light attenuation by plankton biomass and sea ice cover (Equation 12 in Kvale et al., 2021) and is also scaled by temperature and iron availability (Equation 9 in Kvale et al., 2021). In Fig. R2 and R3 we show Hovmöller diagrams showing the seasonal evolution of the proportion of ocean grid cells at each latitude band that are limited by each of these factors for diatoms and coccolithophores in our simulations.

At low iron levels (our 70ka-control run, Fig. R2, added as Figure A9 in the manuscript), light limits the growth of coccolithophores and diatoms south of ~75°S throughout the year (Fig. R2a, c), while nitrate

limits the growth of these plankton functional types in more than 80% of the cells north of ~40°S (Fig. R2b, d). In the band between ~40°S and ~75°S, the limitation moves back and forth between light and nitrate for coccolithophore growth (Fig. R2a, b) and between light, nitrate, and silicate for diatom growth (Fig. R2c-e). Phosphate only limits the plankton growth in the 70ka-control run for a very short time at the northern boundary of our region of interest (~20°S). We therefore concentrate on nitrate and silicate in our analysis.

[Figure]

Fig.R2: Hovmöller diagrams of the proportion of ocean grid cells per latitude band for the 70ka-control experiment for which (a) light is limiting coccolithophore growth (b) nitrate is limiting coccolithophore growth, (c) light is limiting diatom growth (d) nitrate is limiting diatom growth, (e) silicate is limiting diatom growth. In all the subpanels, if shade=1, then all ocean grid cells at that latitude band are limited by the respective limiting factor, if shade=0.5, then half of them are.

To illustrate the changes occurring as more iron is added into the Southern Ocean, we show the changes in the proportion of ocean cells limited by each factor in experiment lambfe20% compared to the 70ka-control (Fig. R3, added as Figure A10 in the manuscript). The region between 40°S and 50°S becomes significantly more nitrate limited for both coccolithophores and diatoms in the lambfe20% experiment (Fig. R3a, b). With higher iron availability, NPP and export production south of 47°S increases, using nutrients more efficiently (Fig. R1c, now added as Figure A9c in the appendix) and reducing the northward nutrient advection (Fig. R1b, now added as Figure A9b in the appendix). At the same time silicate limitation for diatoms decreases in this region (Fig. R3c). This is because, as

the iron flux increases, both coccolithophores and diatoms shift southward leading to a decrease in diatoms between 40°S and 55°S. This further enhances silica availability in this region, consequently leading to a decrease in silicate limitation.

We also show the proportion of ocean cells that are limited by either of the macronutrients (nitrate or phosphate for coccolithophores; and nitrate, phosphate, or silicate for diatoms) in Fig. R3d and e. In this figure we see that the overall nutrient limitation between ~40°S and 50°S increases in the high iron flux experiment compared to 70ka-control.

Therefore, in agreement with the reviewer's comment, we now state that the saturation in the biological pump is indeed due to nitrate limitation in the northern band of the Southern Ocean as the iron fluxes increase.

[Figure]

Fig.R3: Hovmöller diagrams of the lambfe20% minus 70ka-control anomaly in the proportion of ocean grid cells per latitude band for which (a) nitrate is limiting coccolithophore growth, (b) nitrate is limiting diatom growth, (c) silicate is limiting diatom growth, (d) either of the macronutrients is limiting coccolithophore growth and (e) either of the macronutrients is limiting diatom growth.

We now include this information as the following text and as Figures A9 and A10 respectively.

Lines 244-253

However, as the Southern Ocean iron flux increases, the overall iron fertilisation efficiency reduces. While export production increases south of 47°S, thus using nutrients more efficiently (Figure A8c) and reducing the nutrient advection north of the SAF (Figure A8b), nitrate limitation increases in the sub-Antarctic zone (Figure A9b, d and Figure A10a, b), leading to a decrease in export production. For example, in experiment lambfe20%, a 67% decrease in nitrate concentration is simulated between 40°S and 47°S (Figure A8b). At the same time, silicate limitation decreases in the sub-Antarctic zone (Figure 9e and Figure A10c) due to a southward shift of both coccolithophores and diatoms and a decrease in diatoms between 40°S and 55°S (Figure A11). This further enhances silica availability in this region, consequently leading to a decrease in silicate limitation. Therefore, the total biological pump efficiency, represented here by changes in P* (Figure 4c) and Southern Ocean EP to NPP ratio (Figure A8a), saturate at high iron values due to nitrate limitation north of 50°S in the Southern Ocean. The global P* in lambfe20% and in lambfe50%-47S (Figure 4c) equal 0.63 and 0.65 respectively, suggesting a maximum efficiency of 65% in our experimental set-up.

l. 178-182 & 283 – This is not really surprising as EP was likely limited by the scarcity of N (or P) outside of the SO during the last ice age.

We agree with this comment. Here, we are only emphasizing the latitudinal region within the Southern Ocean which is most sensitive to iron fertilisation.

l. 179-188 – this conclusion is at odds with existing paleoceanographic reconstructions. Data suggest that the SAZ was particularly responsive to aeolian Fe supply, while ecosystems in the AZ mainly responded to (micro)nutrient supply from below via upwelling and vertical mixing (e.g. Jaccard et al., 2013). Moreover, N-isotope data suggest that the relative uptake of nitrate increased both in the SAZ (Martinez-Garcia et al., 2014) and the AZ (Studer et al., 2015; Ai et al., 2020) at the MIS4/MIS5 transition.

We thank the reviewer for this comment. Please note that in this section we are discussing differences between two 70 ka BP simulations and not the differences between 70 ka BP and PI. However, we agree with the reviewer's comment that our results are inconsistent with some of the available paleo records. The explanation behind this is now added in the discussion:

Lines 317-339:

Alkenone flux (Lamy et al., 2014; Martinez et al., 2014), as well as opal and organic carbon fluxes (Thöle et al., 2019) reconstructed from subantarctic sediment cores suggest that EP was higher in the SAZ during MIS4 than present day. In addition, bottom water oxygenation records indicate a deep ocean oxygenation decrease during MIS4, which might suggest increased respired carbon storage in the deep ocean (Amsler et al., 2022) but could also reflect a change in ocean dynamics and water residence time. It has been suggested that the subantarctic EP increase during MIS4 was due to higher iron fluxes into the Southern Ocean. Higher dust deposition in the South Pacific and the South Atlantic at 70 ka is consistent with available paleo-proxy records (Martinez et al., 2011, Lambert et al., 2012, Lamy et al.,

2014, Thöle et al., 2019). At the same time, palaeoceanographic records from the AZ suggest a decrease in EP during MIS4 (Anderson et al., 2009; Jaccard et al., 2013; Studer et al., 2015; Thöle et al., 2019; Amsler et al., 2022).

In our 70 ka simulations with enhanced iron input, EP increases in the AZ and polar frontal zone (between SAF and APF) and decreases in the SAZ, in contrast with most paleo-proxy records (Figure A5b). However, there is evidence that greater iron flux within the seasonal sea ice zone has led to higher diatom concentrations (Abelmann et al. [2006, 2015]) and likely also EP in this region during glacial periods. In our study, as the iron input into the ocean increases, the nutrient utilisation increases in the AZ, consistent with the existing $d^{15}N$ records in the AZ at the MIS4/MIS5 transition (Studer et al., 2015; Ai et al., 2020). As a result, the nutrient advection into the SAZ is reduced, thus leading to an EP decrease in the SAZ. In addition, our simulated EP increase in the AZ during MIS4 compared to PI, and thus increase in regenerated carbon storage in the deep ocean, is consistent with some of the proxy records suggesting a decrease in deep ocean oxygenation in the AZ (Jaccard et al., 2016, Amsler et al., 2022). Our EP increase is also consistent with the increase in opal flux north of the APF (Amsler et al., 2022) at the MIS4 onset (Figure A5b).

l. 211 - does the reduction in terrestrial vegetation and soil moisture imply more dust production (positive feedback)?

We thank the reviewer for this interesting comment. This could indeed be positive feedback. However, this feedback is not included in our simulation as the dust flux is a forcing. This would be an interesting topic for future studies.

l. 247 – these observations would be consistent with a general decrease in deep ocean oxygenation at the onset of MIS4 (e.g. Jaccard et al., 2016; Amsler et al., 2022).

We thank the reviewer for pointing this information. It is now included in the discussion as below:

Lines 336-339:

In addition, our simulated EP increase in the AZ during MIS4 compared to PI, and thus increase in regenerated carbon storage in the deep ocean, is consistent with some of the proxy records suggesting a decrease in deep ocean oxygenation in the AZ (Jaccard et al., 2016, Amsler et al., 2022). Our EP increase is also consistent with the increase in opal flux north of the APF (Amsler et al., 2022) at the MIS4 onset (Figure A5b).

---

## Author Response (AR2)

I commend the authors for their careful consideration of the reviewers' comments. I feel the argumentation is more robust in the revised manuscript, yet I feel some (relatively minor) aspects still need to be clarified.

We thank the reviewer for their very insightful comments and are now clarifying our response further.

L. 375-386. I welcome the newly added paragraph, yet I feel the discussion developed in this section remains largely descriptive. The supply of dissolved Fe (and more generally, macronutrients) to the ocean surface in the AZ is to a large extent determined by vertical mixing/upwelling (e.g. Lefevre and Watson, 1999; Tagliabue et al., 2017), possibly explaining why EP decreased in the AZ during past ice ages, if one considers palaeoceanographic reconstructions. Indeed, several authors have argued that EP decreased in the AZ as a result of increased water column stratification as global climate cooled (e.g. Francois et al., 1997; Jaccard et al., 2013; Sigman et al., 2021). How does vertical mixing/upwelling change in the glacial simulations, when compared to the PI timeslice?

We agree that physical mechanisms are important in controlling the magnitude of export production in the AZ. In our simulations, we have kept surface winds prescribed to present day conditions in both the 70ka-control and PI-control simulations, making it difficult to address changes in upwelling due to potential changes in magnitude or latitudinal shifts in westerly winds. With our constant wind forcing, the winter mixed layer in the 70ka-control experiment is slightly deeper compared to the PI-control. This leads to a slight increase in macronutrients in the AZ. Without changes in iron supply (70ka-control experiment), there is an increase in EP in the SAZ and a decrease in EP in the AZ compared to PI (as mentioned in Section 3.1), in agreement with paleo proxy records. When iron supply is increased within this diatom-dominated region, it results in higher utilisation of available nutrients, leading to greater EP within the AZ. We are now including this information in the manuscript as below.

Lines 321- 335:

"In our 70 ka simulations with enhanced iron input, EP increases in the AZ and polar frontal zone (between SAF and APF) and decreases in the SAZ, in contrast with most paleo-proxy records (Figure A5b). Existing $\delta^{15}N$ records suggest higher nutrient consumption at the MIS4/MIS5 transition in the AZ (Studer et al., 2015; Ai et al., 2020), a region where the supply of macronutrients to the surface is determined by ocean upwelling and mixing (Lefevre and Watson, 1999; Tagliabue et al., 2017). Studer et al., (2015) argues that the relative increase in nutrient utilisation in the AZ during MIS4 is due to the general decrease in the nitrate supply, resulting from greater isolation of the deep ocean during glacial periods (Francois et al., 1997; Jaccard et al., 2013; Sigman et al., 2021), rather than iron fertilisation.

This is in contradiction to our results, where we find a deepening of the winter mixed layer in the 70ka simulations compared to the PI-control, which leads to a slight increase in macronutrients in the AZ. Higher iron supply within this diatom-dominated seasonal sea ice zone then leads to a greater utilisation of available nutrients, in agreement with previous studies (Abelmann et al., 2006, 2015), resulting in greater EP within the AZ. With higher nutrient utilisation in the AZ, the nutrient advection into the SAZ is reduced, leading to a decrease in EP in the SAZ. The disagreement between the modelling results and paleo-proxy records thus suggests that the AZ is not stratified enough in our 70ka simulations. This could be due to a weakening and/or more equatorward position of the SH westerly winds during glacial times (e.g., Toggweiler et al., (2006), Gray et al., (2023)), which are not included in our simulations as present-day surface winds are prescribed."

The discussion related to the d15N records Is Insightful, yet, the reasons put forth in the referenced studies are different than the arguments developed here. Indeed, Studer et al., 2015 have argued that the increase in relative nitrate utilization during MIS4, was due to a general decrease in the supply of nitrate via upwelling (and not Fe fertilization). I certainly agree that the cross-frontal (surface) transport of nitrate decreased as a result, yet palaeoceanographic records show no sign of a decrease across MIS4 in the SAZ in all sectors of the SO. Taken together, the palaeoceanographic evidence is consistent with a strengthening of the biological carbon pump, enhancing the storage of remineralised carbon in the ocean interior.

We have now revised our argument concerning the $\delta^{15}N$ records as in the response above and made it clear that the EP response in the SAZ and AZ in our results is inconsistent with most paleo records. As stated in the revised manuscript (Lines 321-324), there is evidence of a greater iron flux in the AZ (Lambert et al., 2015) and evidence of a higher iron flux within the seasonal sea ice zone resulting in increased diatom concentrations (Abelmann et al., 2006, 2015) and likely enhanced EP in this region during glacial periods. Therefore, our results are somewhat consistent with a few of the paleo records.

As such, I would encourage the authors to develop this discussion further. I mean, taken at face value, the model outputs are at odds with palaeoceanographic reconstructions in the entire SO (i.e. both AZ and SAZ) and the North Pacific (which I agree does not warrant more detailed discussions). I do not necessarily argue that palaeoceanographic records are to be taken at face value, yet the fundamental discrepancy between model outputs and paleoproductivity records should be discussed in more details before the manuscript can be accepted for publication.

We agree and have stated the inconsistency in our results with paleo data in the response above. We have also added the following paragraph in addition to the above response.

Lines 336-343:

"However, our simulated EP increase in the AZ during MIS4 compared to PI, and thus the increase in regenerated carbon storage in the deep ocean, aligns with some proxy records suggesting a decrease in deep ocean oxygenation in the AZ (Jaccard et al., 2016; Amsler et al., 2022). Our EP increase is also consistent with the increase in opal flux north of the APF (Amsler et al., 2022) at the MIS4 onset (Figure A5b). In summary, while our EP responses in the SAZ and AZ are inconsistent with most of the paleo records, the strengthening of the biological pump leading to greater storage of deep ocean organic carbon is consistent with palaeoceanographic evidence. Since proxy data for dust flux and EP is limited to only a few marine sites and ice cores, additional paleo-proxy records covering a larger area in the Southern Ocean during MIS4 are needed to better quantify the impact of iron fertilization during the glaciation."